# The lncRNA GATA6-AS epigenetically regulates endothelial gene expression via interaction with LOXL2

Philipp Neumann[1,2], Nicolas Jaé[1,2], Andrea Knau[1], Simone F. Glaser[1,2], Youssef Fouani[1], Oliver Rossbach[3], Marcus Krüger[4], David John[1,2], Albrecht Bindereif[3], Phillip Grote [1,2], Reinier A. Boon [1,2] & Stefanie Dimmeler[1,2]

Impaired or excessive growth of endothelial cells contributes to several diseases. However, the functional involvement of regulatory long non-coding RNAs in these processes is not well defined. Here, we show that the long non-coding antisense transcript of GATA6 (GATA6-AS) interacts with the epigenetic regulator LOXL2 to regulate endothelial gene expression via changes in histone methylation. Using RNA deep sequencing, we find that GATA6-AS is upregulated in endothelial cells during hypoxia. Silencing of GATA6-AS diminishes TGF-β2-induced endothelial–mesenchymal transition in vitro and promotes formation of blood vessels in mice. We identify LOXL2, known to remove activating H3K4me3 chromatin marks, as a GATA6-AS-associated protein, and reveal a set of angiogenesis-related genes that are inversely regulated by LOXL2 and GATA6-AS silencing. As GATA6-AS silencing reduces H3K4me3 methylation of two of these genes, periostin and cyclooxygenase-2, we conclude that GATA6-AS acts as negative regulator of nuclear LOXL2 function.

[1] Institute for Cardiovascular Regeneration, Goethe University, Theodor-Stern-Kai 7, Frankfurt am Main 60590, Germany. [2] German Center of Cardiovascular Research (DZHK), Frankfurt am Main 60590, Germany. [3] Institute of Biochemistry, Justus-Liebig-University, Heinrich-Buff-Ring 17, Giessen 35392, Germany. [4] Max Planck Institute for Heart and Lung Research, Ludwigstraße 43, Bad Nauheim 61231, Germany. Philipp Neumann and Nicolas Jaé contributed equally to this work. Correspondence and requests for materials should be addressed to S.D. (email: dimmeler@em.uni-frankfurt.de)

High-throughput sequencing-based profiling of 15 different cell lines revealed that ~74% of the human genome is transcribed, however, only ~2% actually account for protein-coding genes[1,2]. As a consequence, the majority of the human transcriptome can be referred to as non-coding RNA. According to their size, non-coding RNAs are subdivided into small non-coding RNAs (<200nt) and long non-coding RNAs (lncRNAs; >200nt); the latter class being mainly unannotated and uncharacterized[3]. On a functional level, lncRNAs are implicated in complex biological processes through diverse mechanisms. These comprise, among others, gene regulation by titration of transcription factors, splicing alteration, sponging of microRNAs and recruitment of chromatin modifying enzymes[4-7]. For example, recent studies suggest that the intergenic lncRNA H19 interacts with methyl-CpG-binding domain protein 1 to recruit H3K9 methyltransferases to its own imprinted gene network[8]. Beyond being functionally restricted to their own site of transcription (*cis*-regulation), lncRNAs can also regulate gene expression on distant chromosomal sites (*trans*-regulation). This mode of action is for example known for the lncRNA GAS5, which is transcribed from chromosome 1, however, by acting as a ribo-repressor of the glucocorticoid receptor, inhibits transcription of distinct genes located on various chromosomes[9].

Regarding the cardiovascular field, several recent studies identified lncRNAs that control endothelial cell functions[10]. Silencing or genetic deletion of the hypoxia-induced lncRNA MALAT1 impairs the expression of various cell cycle regulators and interferes with endothelial cell proliferation, migration, postnatal retina vascularization and neovascularization after hind limb ischemia[11,12]. Moreover, hypoxia was shown to regulate the long non-coding RNAs LINC00323 and MIR503HG and silencing of these transcripts led to angiogenic defects in vitro[13]. Another vascular enriched long non-coding RNA, SENCR, was found to control the angiogenic capacity of human umbilical vein endothelial cells (HUVECs) by specifically regulating endothelial gene expression[14]. Endothelial-specific functions were also described for the non-coding RNAs tie-1AS, which overlaps with and selectively binds to tie-1 mRNA[15] as well as for Meg3, which is supposed to epigenetically regulate endothelial gene expression[16,17]. Additionally, the lncRNAs TERMINATOR, ALIEN, and PUNISHER were found to be induced during differentiation and act as important contributors to endothelial commitment and identity[18]. However, the molecular mechanisms by which these lncRNAs regulate endothelial cell functions and neovascularization mainly remain elusive.

In this study, we determine the influence of hypoxia on lncRNA expression in HUVECs, and provide mechanistic insights into the epigenetic regulation of endothelial gene expression and angiogenesis by the interplay between the chromatin modifying enzyme LOXL2 and its negative regulator, the lncRNA GATA6-AS.

## Results

**Hypoxia-induced GATA6-AS controls endothelial cell functions**. To analyze the influence of hypoxia on lncRNA expression in HUVECs, we performed deep sequencing of ribo-minus RNA isolated from cells that were exposed to hypoxic (12 and 24 h, 0.2% $O_2$) or normoxic conditions. After filtering data for long non-coding RNA annotation, expression levels, and significant response to hypoxia, we identified 54 transcripts to be upregulated and 10 to be downregulated upon 24 h of hypoxia (Fig. 1a, Supplementary Table 1). Besides previously described hypoxia-sensitive lncRNAs, like MALAT1[11] or the miR210 host gene[19], we additionally identified numerous poorly characterized or uncharacterized transcripts. Among the latter ones, RP11-

627G18.3 –also annotated as GATA6-AS– showed a strong as well as consistent response to hypoxia after 12 to 24 h of incubation, similar to the known hypoxia response of VEGFA mRNA (Fig. 1b). In order to validate the expression and regulation of GATA6-AS, we performed RT-qPCR using two different primer pairs and confirmed GATA6-AS to be ~2.5 fold upregulated after 24 h of hypoxia, compared to normoxic control conditions (Fig. 1c, Supplementary Fig. 1a, b). Of note, induction of GATA6-AS was maintained by prolonged hypoxic incubation (Supplementary Fig. 1c). By analyzing publicly available transcription factor chromatin immunoprecipitation (ChIP)-seq data from HUVECs and in silico prediction, we found that the GATA6-AS promotor region harbors binding sites for the hypoxia-induced transcription factor HIF1α, as well as for E2F1 and EGR1 (Supplementary Fig. 1d), which are implicated in the hypoxia-controlled PI3K/Akt signaling cascade[20-22]. Whereas silencing of HIF1α did not affect GATA6-AS expression (Supplementary Fig. 1e), pharmacological inhibition of Akt significantly reduced hypoxia-driven GATA6-AS induction (Fig. 1e, Supplementary Fig. 1f). Next, we determined the subcellular localization of this lncRNA in HUVECs by cellular fractionation. Interestingly, GATA6-AS predominantly localized to the nucleus, whereas protein-coding mRNAs, e.g. RPLP0, were found in the cytoplasm (Fig. 1d). This subcellular distribution was not significantly changed under hypoxic conditions (Supplementary Fig. 1g). The predominant nuclear localization implies a non-protein-coding nature for GATA6-AS and is in agreement with our in silico analysis of all annotated GATA6-AS transcript variants by a coding potential assessment tool[23] which revealed coding probabilities significantly smaller than the default cut off for protein-coding human transcripts (Supplementary Table 2).

Next, we determined the function of GATA6-AS in endothelial cells in vitro. Since hypoxia induces GATA6-AS, we first analyzed the role of GATA6-AS in the cellular mechanism of endothelial-mesenchymal transition (EndMT), a process promoted by hypoxia and recently being recognized as causal contributor of various cardiovascular pathologies, including tissue fibrosis after injury[24,25]. To this end, GATA6-AS expression was silenced in HUVECs by transfecting locked nucleic acid (LNA) GapmeRs (Fig. 2a, Supplementary Fig. 1b) and the effects of silencing were assessed in TGF-ß2-induced EndMT assays. Whereas TGF-ß2 treatment strongly induced EndMT, as documented by an increased expression of the mesenchymal markers SM22 and calponin and a simultaneous reduction of VE-cadherin protein levels at the cell junctions, silencing of GATA6-AS diminished these effects and prevented EndMT (Fig. 2b, c). In parallel, we assessed angiogenic sprouting in vitro by spheroid assays, which mainly determines the tip cell formation capacity of endothelial cells. Here, GATA6-AS silencing reduced spheroid outgrowth by about 50 and 28% under basal conditions and VEGF stimulation, respectively (Fig. 2d, Supplementary Fig. 2a, b). Consistently, migration of endothelial cells in transwell assays was significantly reduced by GATA6-AS silencing (Fig. 2e). In contrast, endothelial cell proliferation and apoptosis were not significantly affected (Supplementary Fig. 2c, d). Together, these data show that GATA6-AS silencing controls endothelial-mesenchymal transition, which may lead to impaired in vitro sprouting and endothelial cell migration.

Next, we aimed to address the relevance of these features in vivo. Due to lack of sequence conservation in mice, we explored the expression of transcripts at the Gata6 locus. Even though, several putative murine transcripts are annotated (Supplementary Fig. 3a), transcripts could only be detected during mouse development, but not in cultured adult mouse endothelial cells (Supplementary Fig. 3b, c). Gata6-AS was also not detectable in ischemic and non-ischemic adult mouse

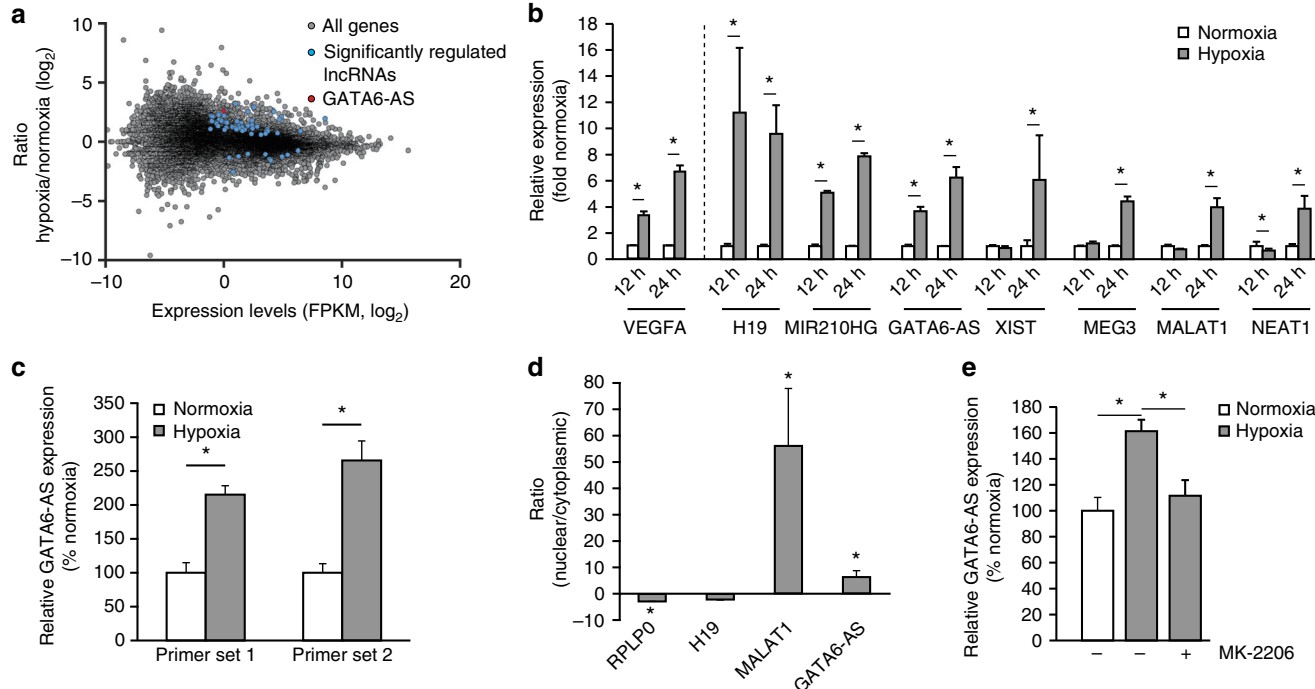

**Fig. 1** The lncRNA GATA6-AS is hypoxia-induced and nuclear enriched in endothelial cells. **a**, **b** HUVECs were exposed to hypoxia (0.2% $O_2$) for 12 and 24 h or kept under normoxic conditions and gene expression changes were assayed by deep sequencing of ribo-minus RNA. **a** Significantly regulated lncRNAs are highlighted in blue, GATA6-AS is highlighted in red ($n = 2$; statistical significance was assessed by Cuffdiff). **b** FPKM values were used to identify significantly regulated lncRNAs. VEGFA is shown as hypoxia-induced positive control ($n = 2$; SEM; *statistical significance was assessed by Cuffdiff). **c** HUVECs were exposed to hypoxia (0.2% $O_2$) for 24 h or kept under normoxic conditions and regulation of GATA6-AS was confirmed by RT-qPCR using two primer sets targeting different regions of GATA6-AS. Relative expression was normalized to RPLP0 mRNA ($n = 4$; SEM; * t test $p < 0.05$). **d** RNA was isolated from nuclear and cytoplasmic fractions and analyzed by RT-qPCR using primers targeting the indicated transcripts ($n = 4$; SEM; * t test $p < 0.05$). **e** Akt phosphorylation was inhibited in HUVECs by addition of MK-2206 to the culture medium 1 h prior to hypoxic treatment (24 h, 0.2% $O_2$). As controls, cells were incubated under normoxic conditions. Expression levels of GATA6-AS were determined by RT-qPCR and normalized to RPLP0 mRNA ($n = 4$; SEM; * t test $p < 0.05$)

tissue (Supplementary Fig. 3d), implicating that the function of Gata6-AS in the neovascularization response after hind limb ischemia cannot be studied. To circumvent this limitation, we used a human endothelial cell-based xenograft model[26,27], in which HUVECs, transfected with control GapmeRs or GapmeRs directed against GATA6-AS were transplanted in immune deficient mice. Histological analysis revealed that GATA6-AS-silenced human cells formed significantly more vessels in comparison to controls, whereas endogenous murine vessels were similar in both groups (Fig. 2f–h). Moreover, vessels formed by GATA6-AS-silenced endothelial cells were perfused as evidenced by the detection of erythrocytes in the lumen, indicating that these vessels are functionally mature (Supplementary Fig. 2e). Together, these data demonstrate that GATA6-AS silencing promotes vessel formation in vivo.

**GATA6-AS does not act in a *cis*-regulatory manner.** Since classical antisense transcripts may preferentially control the expression of the corresponding sense transcripts[28–30], a mode of action also suggested for the putative murine homolog of GATA6-AS during early embryonic development[31], we analyzed GATA6 mRNA and protein expression levels in our experimental setup. However, a *cis*-regulatory function of GATA6-AS on the adjacent transcription factor GATA6 itself was not detectable in GATA6-AS-silenced normoxic or hypoxic endothelial cells (Supplementary Fig. 4a, b). Consequently, GATA6 target gene expression, e.g. TEK, EDN1 and SERPINE1, was also not affected by silencing of GATA6-AS (Supplementary Fig. 4c).

**GATA6-AS interacts with the lysyl oxidase LOXL2.** After ruling out a *cis*-regulatory function for GATA6-AS in endothelial cells, we aimed to identify putative protein binding partners of this lncRNA by antisense affinity selection to mechanistically characterize the observed biological phenotypes. Therefore, we first used RNase H-based cleavage of RNA-DNA-heteroduplexes and subsequent RT-qPCR to determine oligonucleotide-accessible regions within the GATA6-AS transcript (Fig. 3a, b, Supplementary Fig. 1b). Subsequently, the accessible sequence of antisense oligo #1 (AS1) was used to design a 2′O-Me-RNA antisense probe carrying a 3′-desthiobiotin-TEG group for streptavidin pulldown of endogenous GATA6-AS-protein complexes under native conditions (Fig. 3c, Supplementary Fig. 1b). Using this probe in HeLa cell lysates, in which GATA6-AS was similarly induced by hypoxia as in endothelial cells (Supplementary Fig. 5a, b), we were able to specifically enrich for GATA6-AS, as seen by RT-qPCR on the RNA fraction of the affinity selection (Fig. 3d). In parallel, biotin eluted protein fractions were separated by SDS-PAGE (Supplementary Fig. 5c), analyzed by mass spectrometry, and identified proteins were sorted according to their enrichment over control selections (Supplementary Table 3). Among the top enriched proteins, we identified lysyl oxidase-like 2 (LOXL2) as a putative GATA6-AS-binding protein, which has established functions in the hypoxic response of endothelial cells as well as in the process of angiogenesis[32]. To confirm the GATA6-AS-LOXL2 interaction, we first repeated the affinity selection of GATA6-AS in HeLa cell lysates, and showed LOXL2 association by subsequent western blot analysis of co-precipitated proteins (Fig. 4a). Second, we performed RNA immunoprecipitation (RIP) in

HUVEC lysates, using antibodies directed against LOXL2 and analyzed co-precipitated RNAs by RT-qPCR (Fig. 4b). Of note, LOXL2-RIP resulted in a specific enrichment of co-precipitated GATA6-AS, compared to isotype control reactions and hypoxic conditions further augmented GATA6-AS enrichment (Supplementary Fig. 5d). Third, we used recombinant His-LOXL2 and in vitro transcribed full length GATA6-AS in His-tag pulldown reactions to subsequently assay for GATA6-AS binding by Northern blotting (Fig. 4c). Whereas His-tagged LOXL2 clearly

enriched for GATA6-AS, mock pulldown reactions showed no GATA6-AS enrichment. Fourth, to definitely show the direct interaction between GATA6-AS and LOXL2 under physiological conditions, we performed individual nucleotide resolution crosslinking and immunoprecipitation (iCLIP) of endogenous LOXL2 in HUVECs (Fig. 4d, top, Supplementary Fig. 5e). RNA deep sequencing revealed two LOXL2 iCLIP-tags on GATA6-AS (Fig. 4d, bottom), but none in control reactions. Having shown that GATA6-AS directly interacts with LOXL2, we sought to

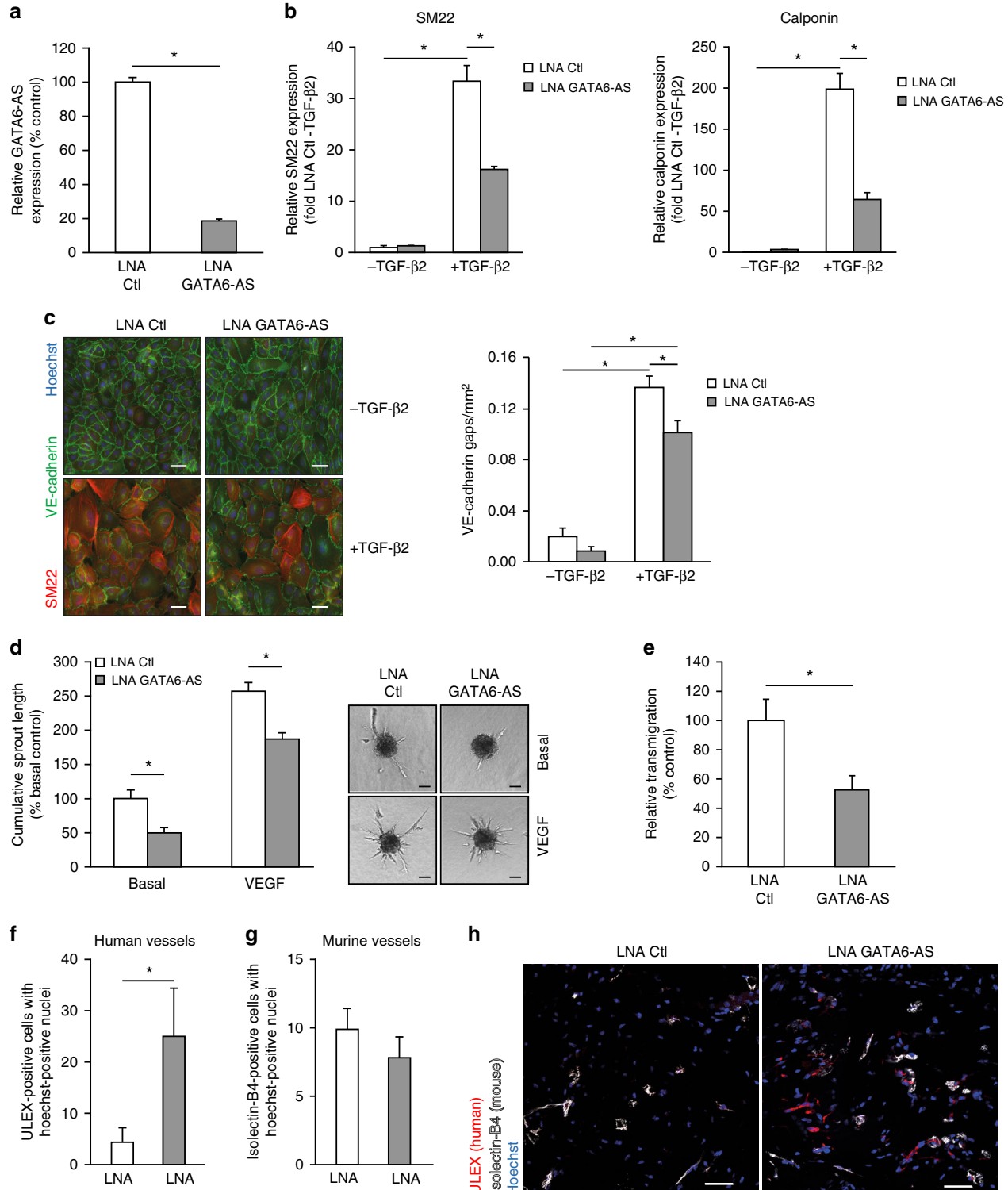

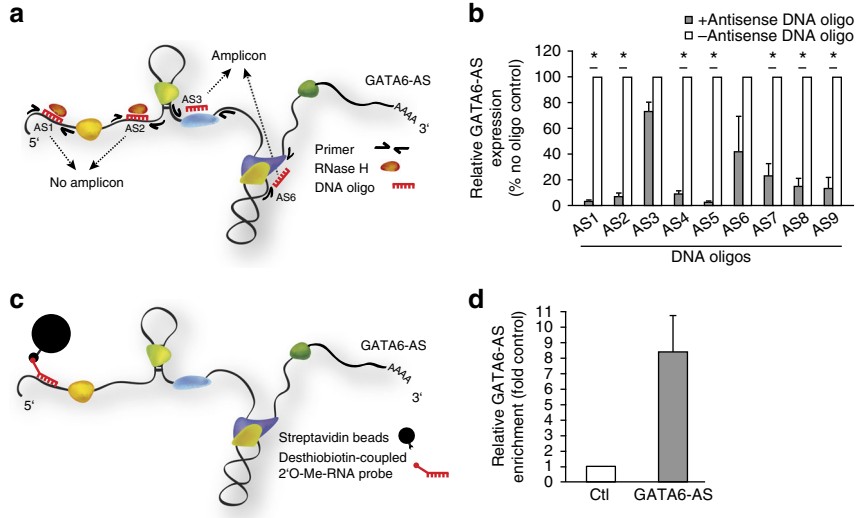

**Fig. 3** Antisense affinity selection of endogenous GATA6-AS–protein complexes. **a** Scheme for testing oligonucleotide accessibility of GATA6-AS. Binding of antisense DNA oligonucleotides to accessible sites within the GATA6-AS transcript results in the formation of DNA-RNA-heteroduplexes and subsequent RNA cleavage by RNase H. Cleaved sites cannot be amplified by flanking primers in subsequent RT-qPCR reactions. **b** DNA oligonucleotides (AS1–9; see Supplementary Fig. 1b for details) targeting GATA6-AS were incubated in HUVEC cellular extract and RNase H was added to degrade DNA-RNA-heteroduplexes. Following RNA preparation, RT-qPCR was used to assay for accessible sites. In control reactions, DNA oligonucleotides were omitted ($n = 3$; SEM; *$t$ test $p < 0.05$). **c** Scheme for purifying endogenous GATA6-AS–protein complexes. An oligonucleotide-accessible site within GATA6-AS is targeted by a desthiobiotin-coupled 2′O-Me-RNA probe. Subsequently, the complex is recovered using streptavidin beads. **d** HeLa cell lysate was used to capture endogenous GATA6-AS–protein complexes by RNA affinity selection (AS1 probe). As control, scrambled 2′O-Me-RNA probes were used. Following elution with biotin, co-purified RNA fractions were analyzed for GATA6-AS enrichment by RT-qPCR ($n = 4$; SEM)

characterize this interaction in depth. Since LOXL2 is not known to be a classical RNA-binding protein, we analyzed whether RNA structure might be determinative for the observed LOXL2 interaction. To this end, we folded in vitro transcribed full length GATA6-AS in a multi-step incubation process, or-as an alternative-denatured the same transcript at 95 °C and used these pre-treated RNAs in His-LOXL2 pulldown assays (Fig. 4e). Interestingly, folding of GATA6-AS augmented binding to LOXL2, whereas denaturation of the same transcript reduced LOXL2 association to levels observed in mock pulldown controls. Moreover, we compared LOXL2 binding of folded full length GATA6-AS to a folded mutant transcript lacking the surrounding sequences of the identified LOXL2-binding sites (Fig. 4f). The mutant GATA6-AS transcript exhibited strongly reduced binding affinities, comparable to those obtained from full length GATA6-AS mock controls. In summary, these results establish LOXL2 as direct binding partner of the lncRNA GATA6-AS and place a

special emphasis on the important role of RNA structure in this RNA-protein-interaction.

**GATA6-AS negatively regulates nuclear LOXL2 function.** LOXL2 mediates the oxidative deamination of lysine residues on target proteins leading to the formation of allysine, a process implicated in numerous cellular and extracellular processes[33,34]. For instance, it is known that LOXL2, when secreted into the extracellular matrix, regulates angiogenesis through collagen IV scaffolding[32]. Therefore, we silenced LOXL2 expression in endothelial cells (Fig. 5a, b) and assayed for the resulting sprouting phenotype (Fig. 5c). Interestingly, silencing of LOXL2 significantly reduced the outgrowth of sprouts, compared to control conditions. Since the angiogenic capacity of LOXL2 is attributed to its secreted form, we analyzed intracellular and extracellular levels of LOXL2 upon silencing of GATA6-AS or LOXL2 (Fig. 5d). As expected, silencing of LOXL2 decreased

**Fig. 2** GATA6-AS silencing reduces EndMT in vitro and augments vessel formation in a xenograft model. **a** HUVECs were transfected with GapmeRs targeting GATA6-AS or with control GapmeRs and relative expression of GATA6-AS was determined by RT-qPCR, normalized to RPLP0 mRNA ($n = 4$; SEM; *$t$ test $p < 0.05$). **b**, **c** GATA6-AS-silenced cells or control cells were cultivated for 72 h in complete medium (−TGF-β2) or medium lacking endothelial growth factor and bovine brain extract, but containing 10 ng/μl TGF-β2 (+TGF-β2). **b** Relative expression levels of SM22 (left) and calponin (right) were determined by RT-qPCR, normalized to RPLP0 mRNA ($n = 2$–3; SEM; *$t$ test $p < 0.05$). **c** Left: Cells were immunostained for SM22 (red) and VE-cadherin (green) and nuclei were counter stained using Hoechst (representative images are shown; scale bars are 50 μm). Right: For quantification, VE-cadherin gaps per mm² were determined in 9–10 random fields per condition ($n = 5$; SEM; *$t$ test $p < 0.05$). **d** HUVECs were transfected with GapmeRs targeting GATA6-AS or with control GapmeRs and used for in vitro spheroid sprouting assays under basal conditions and VEGFA (50 ng/ml) stimulation ($n = 4$; SEM; *$t$ test $p < 0.05$; representative images are shown; scale bars are 50 μm). **e** GATA6-AS-silenced cells or control cells were used for in vitro transmigration assays under VEGFA (50 ng/ml) stimulation ($n = 6$; SEM; *$t$ test $p < 0.05$). **f**–**h** Matrigel basement matrix plugs containing spheroids derived from GATA6-AS or control GapmeR treated HUVECs were injected subcutaneously in immunodeficient mice. Plugs were harvested 21 days later and used for immunohistochemistry detecting human endothelial cells (ulex rhodamine, red) and murine endothelial cells (isolectin-B4, white). Nuclei were stained with Hoechst (blue). **f** Implanted human endothelial cells were quantified as cells being ulex-posistive and Hoechst-positive ($n = 10$ plugs; SEM; *Mann–Whitney $U$ test $p < 0.05$). **g** Murine endothelial cells were quantified as cells being isolectin B4-positive and Hoechst-positive ($n = 10$ plugs; SEM). **h** Representative images of GATA6-AS and control GapmeR matrigel plug sections. Scale bars are 50 μm

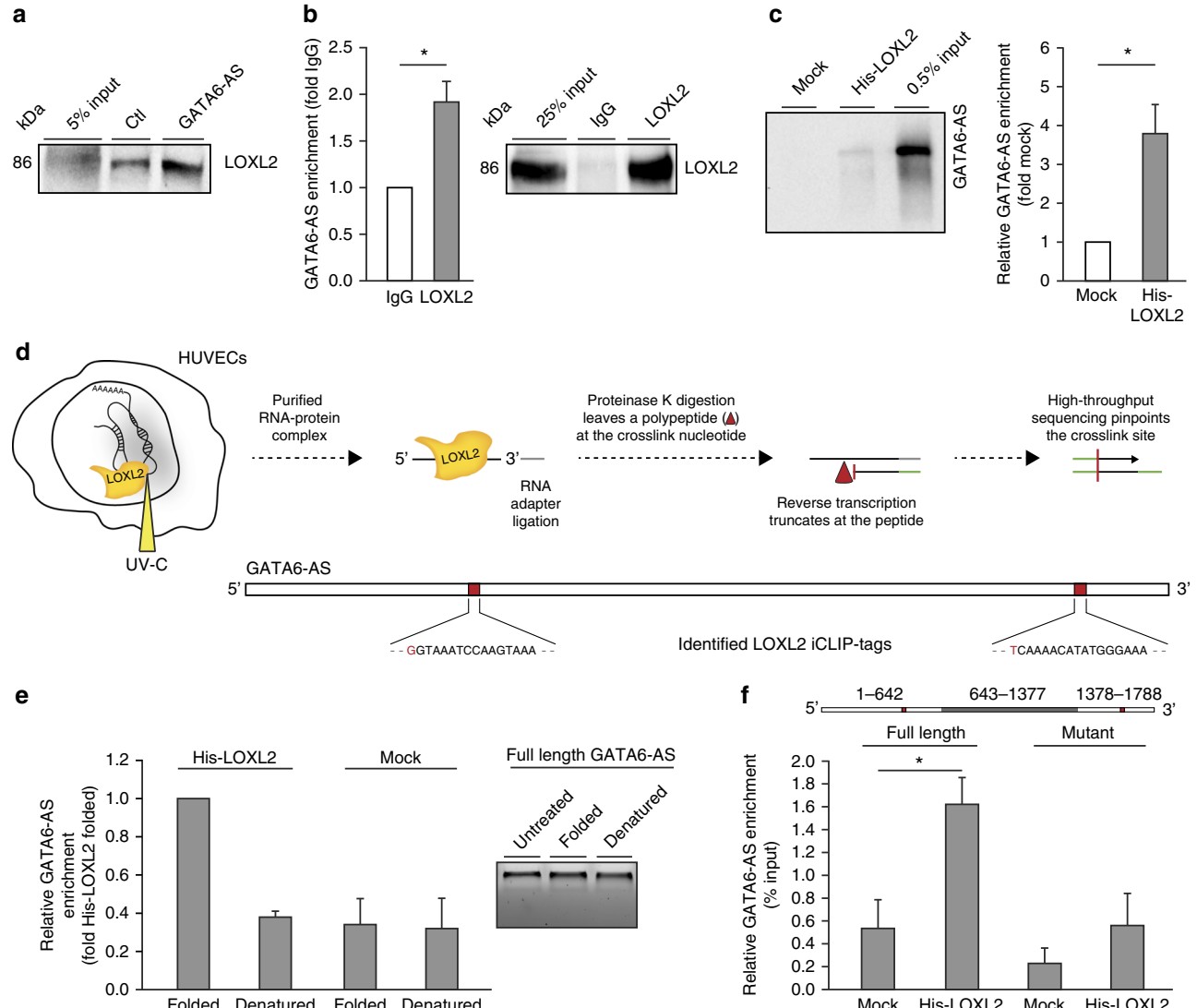

**Fig. 4** GATA6-AS is directly bound by LOXL2. **a** Total cell lysate was prepared from HeLa cells and 5% was taken as input. The remaining lysate was distributed between GATA6-AS or scramble control antisense affinity selections and co-purified proteins were assayed for LOXL2 by western blotting ($n$ = 3; a representative image is shown). **b** HUVEC cell lysate was used for RNA immunoprecipitation using LOXL2 antibodies or isotype controls. Left: Co-purified RNA from LOXL2 and IgG IPs was assayed for GATA6-AS enrichment by RT-qPCR ($n$ = 5; SEM; * $t$ test $p$ < 0.05). Right: IP specificity was controlled by LOXL2 western blotting (a representative image is shown). **c** In vitro transcribed full length GATA6-AS (1788nt) was used in His-LOXL2 pulldown assays. As control, mock pulldowns were performed without protein. Left: Co-purified GATAT6-AS was visualized by Northern blotting (a representative image is shown). Right: For quantification, intensities were determined by densitometry ($n$ = 4; SEM; *$t$ test $p$ < 0.05). **d** Schematic outline of the LOXL2 iCLIP procedure in HUVECs (top) and identified LOXL2-binding sites within GATA6-AS (bottom). The crosslinked nucleotides of the LOXL2 iCLIP-tags are highlighted in red. **e** Full length GATA6-AS (1788nt) was in vitro transcribed and folded (5 min 65 °C → 10 min RT → 30 min 30 °C → 15 min 37 °C) or denatured (2 min 95 °C → ice) and used for His-tag pulldown using recombinant LOXL2. As controls, mock reactions were performed without protein. Left: Co-purified RNA was recovered and used for RT-qPCR ($n$ = 4; SEM). Right: Integrity of GATA6-AS input after folding or denaturation was assayed by agarose gel electrophoresis (a representative image is shown). **f** Full length GATA6-AS and a 735nt truncated construct (nucleotides 643–1377) lacking both LOXL2 interaction sites were in vitro transcribed, folded, and used for His-LOXL2 pulldown. As controls, mock reactions were performed without protein. Co-purified RNA was recovered and used for RT-qPCR ($n$ = 4; SEM; *$t$ test $p$ < 0.05)

intra- and extracellular LOXL2 protein levels, however, LOXL2 levels remained unchanged upon silencing of GATA6-AS. In order to analyze the role of extracellular LOXL2 in sprouting angiogenesis in more detail, we silenced LOXL2 in endothelial cells and determined whether treatment with supernatant from control-treated cells restored sprouting behavior. Indeed, supernatants of control cells, but not from siLOXL2 silenced endothelial cells rescued the angiogenesis defect in LOXL2 siRNA transfected cells (Supplementary Fig. 6). These results, together

with the demonstrated nuclear localization of GATA6-AS, suggest that GATA6-AS silencing does not alter extracellular LOXL2 levels and that the LOXL2 sprouting phenotype is driven by an extracellular depletion of LOXL2. Nuclear LOXL2, in contrast, is known to act as a transcription co-repressor by specifically deaminating trimethylated lysine 4 of histone H3 (H3K4me3), a chromatin signature for transcriptional activation[35,36]. Thus, we hypothesized that GATA6-AS-as it is predominantly localized in the nucleus-(Fig. 1d, Supplementary Fig. 1g) may regulate nuclear

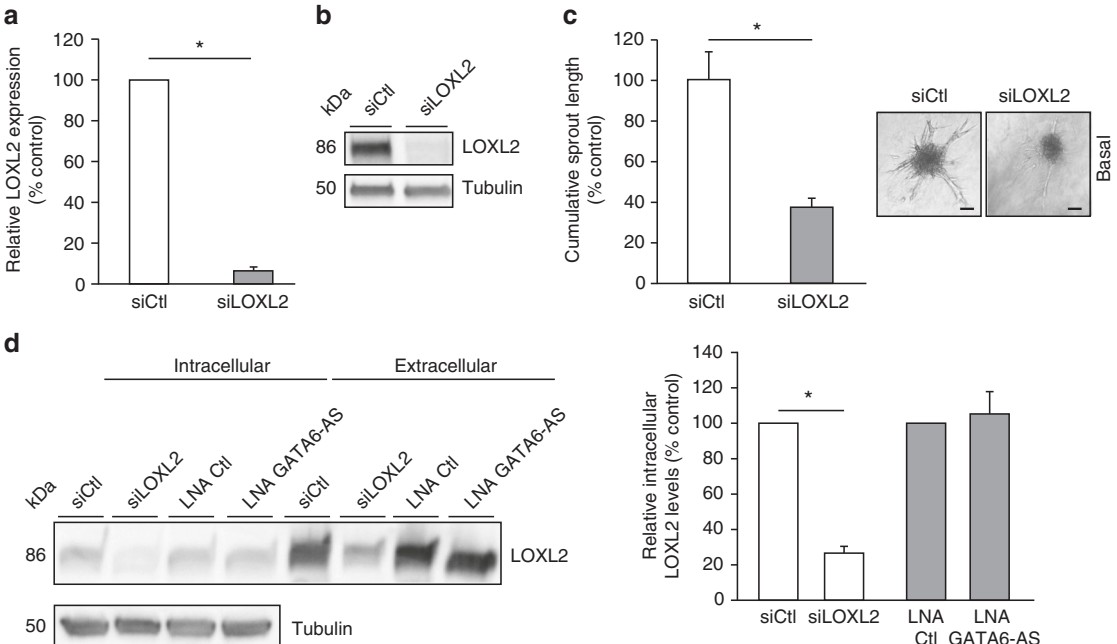

**Fig. 5** GATA6-AS silencing does not influence intracellular and extracellular LOXL2 levels. **a** HUVECs were transfected with siRNAs targeting LOXL2 or with control siRNAs and relative expression of LOXL2 was determined by RT-qPCR, normalized to RPLP0 mRNA ($n = 6$; SEM; *t test $p < 0.05$). **b** LOXL2 protein levels were determined from the same cells by western blotting, using tubulin as loading control ($n = 1$). **c** LOXL2-silenced cells or control cells were used for in vitro spheroid sprouting assays under basal conditions ($n = 6$; SEM; * t test $p < 0.05$; representative images are shown; scale bars are 50 μm). **d** HUVECs were transfected with siRNAs or GapmeRs targeting LOXL2 or GATA6-AS, respectively. Left: Intracellular and extracellular LOXL2 levels were assayed by western blotting (representative images are shown). Right: For quantification, intracellular LOXL2 levels were normalized to tubulin ($n = 5$–6; SEM; *t test $p < 0.05$)

LOXL2 functions thereby controlling endothelial gene expression. To examine this hypothesis, we first controlled for the subcellular distribution of LOXL2 in HUVECs under normoxic and hypoxic conditions and found LOXL2 to localize to both compartments, cytoplasm and nucleus (Supplementary Fig. 7a). Based on these findings, we determined the expression of global H3K4me3 marks in HUVECs and showed that silencing of GATA6-AS significantly reduced H3K4me3 levels by about 30% (Fig. 6a), whereas silencing of LOXL2 augmented H3K4me3 marks (Fig. 6b). Analysis of global H3K4me3 levels under hypoxia confirmed these results (Supplementary Fig. 7b, c). To determine whether the modulation of H3K4me3 levels may result in a change of endothelial gene expression, we used GapmeRs and siRNAs to silence the expression of GATA6-AS or LOXL2, respectively (Supplementary Fig. 7d, e), and identified regulated genes using exon arrays (Supplementary Fig. 7f, g). Interestingly, ~71% of all GATA6-AS-regulated genes (≥1.2 fold upregulated or downregulated) were found to be inversely regulated upon silencing of LOXL2 (Fig. 6c). In detail, 49.8% of genes repressed by GATA6-AS silencing were found to be upregulated after LOXL2 silencing, whereas only 20.9% of upregulated genes were downregulated by LOXL2 silencing (Fig. 6c). Intriguingly, gene ontology analysis of these shared sets of inversely regulated genes revealed a number of transcripts associated with hypoxia and angiogenesis (Supplementary Table 4) and we were able to confirm the inverse regulation for selected candidates by RT-qPCR (Fig. 6d). Based on pronounced H3K4me3 promoter marks (Supplementary Fig. 7h), we selected two inversely regulated genes for a detailed H3K4me3 ChIP-PCR analysis: cyclooxygenase-2 (also known as prostaglandin-endoperoxide synthase 2; PTGS2), which controls endothelial cell phenotypes[37], and the extracellular protein periostin (POSTN), which increases epithelial-mesenchymal transition[38] and promotes angiogenesis via activation of Erk/VEGF signaling[39]. Strikingly, silencing of

GATA6-AS significantly reduced H3K4me3 ChIP efficiencies on the corresponding promoter regions, compared to non-silenced conditions (Fig. 6e). Taken together, these findings provide evidence that the hypoxia-responsive lncRNA GATA6-AS epigenetically regulates endothelial gene expression and angiogenic activity by binding to and interfering with a fraction of the nuclear oxidative deaminase LOXL2.

## Discussion

In this study we identified novel hypoxia-regulated long noncoding RNAs in endothelial cells and elucidated the functional significance and molecular mode of action for the lncRNA GATA6-AS. An initial RNA deep sequencing-based screening revealed hypoxia as a powerful regulator of lncRNA expression and uncovered-in addition to previously annotated endothelial lncRNAs[40]-GATA6-AS to be robustly and consistently upregulated. While several lncRNAs have been identified and described in the cardiovascular system[10], little is known about the detailed molecular roles of these RNAs in vascular biology. Besides the well-studied functions of microRNAs[41], distinct novel noncoding RNA species-e.g. circular RNAs-were recently shown to be hypoxia-regulated and to have functional implications in the endothelium[42]. Nevertheless, our functional and mechanistic understanding of these transcripts is grossly incomplete. Since GATA6-AS is assigned as antisense transcript of the transcription factor GATA6, we first assayed for a putative *cis*-regulatory function. This long-known mode of action is for example described for other antisense transcripts, like the non-coding antisense RNA in the tie-1 locus (tie-1AS) or Dll4-AS, which are implicated in the regulation of the key angiogenesis genes tie-1 and Delta-like 4, respectively[15,43]. However, GATA6-AS silencing did neither influence GATA6 expression itself, nor the expression of GATA6 downstream genes, suggesting that a *cis*-regulatory

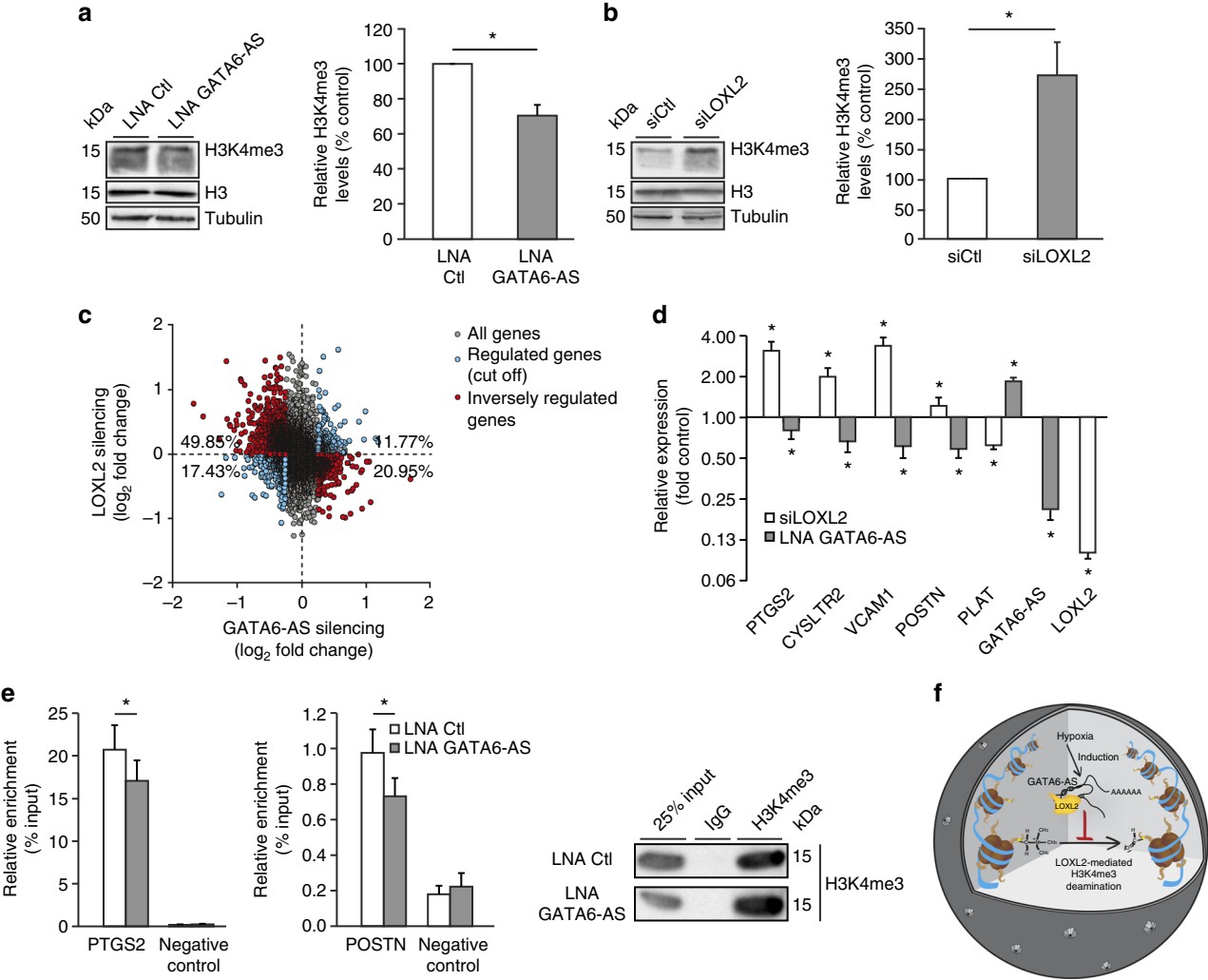

**Fig. 6** GATA6-AS acts as negative regulator of nuclear LOXL2 function in endothelial gene expression. **a, b** Left: HUVECs were transfected with GapmeRs or siRNAs targeting GATA6-AS or LOXL2, respectively and H3K4me3, H3 as well as tubulin levels were assayed by western blotting (representative images are shown). Right: For quantification, H3K4me3 levels were normalized to tubulin ($n = 3$–6; SEM; *$t$ test $p < 0.05$). **c** Upon silencing of GATA6-AS, changes in global gene expression levels were determined by exon arrays (all genes; gray). Regulated genes ($\geq 1.2$ fold upregulated or downregulated) were analyzed in corresponding LOXL2-silenced samples (cut off; blue) and when found inversely regulated, highlighted in red. Percentages refer to the total of all cut off genes. Six data points are outside the axis limits ($n = 2$–4). **d** HUVECs were transfected with siRNAs or GapmeRs targeting LOXL2 or GATA6-AS, respectively and expression of LOXL2 and GATA6-AS, as well as of the indicated angiogenesis-related genes was determined by RT-qPCR, normalized to RPLP0 mRNA ($n = 7$–8; SEM; *$t$ test $p < 0.05$). **e** HUVECs were transfected with GapmeRs targeting GATA6-AS or with control GapmeRs and used for H3K4me3 chromatin immunoprecipitations. Left: Co-precipitated DNA was assayed for PTGS2 and POSTN enrichment by qPCR. A genomic region lacking H3K4me3 signals was used as negative control ($n = 4$–10; SEM; *$t$ test $p < 0.05$). Right: Specificity of the immunoprecipitation was controlled by isotype controls and H3K4me3 western blotting (representative images are shown). **f** Schematic model of GATA6-AS-regulated LOXL2 function. In the nucleus, LOXL2 catalyzes the oxidative deamination of activating H3K4me3 chromatin marks, a process which is negatively regulated by the hypoxia-induced nuclear lncRNA GATA6-AS

function is unlikely to occur in HUVECs. Moreover, analysis of microRNA binding to GATA6-AS by Ago HITS-CLIP (high-throughput sequencing of RNA isolated by crosslinking and immunoprecipitation) uncovered that GATA6-AS is most probably not involved in the process of microRNA sponging (data not shown). Therefore, we aimed to directly identify putative protein binding partners of GATA6-AS. In order to minimize the error-proneness of this approach, we purified and eluted endogenous GATA6-AS–protein complexes by RNA antisense affinity selection under native conditions, a strategy frequently used to purify ribonucleoprotein particles[44], and uncovered the enzyme LOXL2 as GATA6-AS-interacting protein. LOXL2 belongs to the lysyl oxidase family which comprises $Cu^{2+}$-dependent and lysine tyrosylquinone-dependent amine

oxidases[33] and catalyzes the oxidative deamination of the ε-amino group of lysines and hydroxylysines, a process which is well-studied in extracellular matrix remodeling[45–47]. A recent study by Millanes-Romero et al.[35] revealed that this catalytic activity of LOXL2 is additionally linked to transcriptional regulation by deaminating H3K4me3. H3K4 trimethylation is associated with transcriptional activation[48] and demethylation, which is accompanied by transcriptional silencing, is usually achieved by amino-oxidation or hydroxylation leading to the formation of H3K4me2 and H3K4me1 marks[49–52]. LOXL2-catalyzed deamination, however, results in the generation of non-methylated H3K4, representing an unconventional and sparsely characterized way of epigenetic gene silencing. Regarding the functional contribution of GATA6-AS to this process, we confirmed the

interaction between LOXL2 and GATA6-AS by affinity selection, RIP, in vitro His-tag pulldown, and iCLIP. Of note, LOXL2 has not been described as an RNA-binding protein so far and is lacking classical RNA recognition or binding domains[53]. Instead, LOXL2 features four scavenger receptor cysteine-rich domains and a C-terminal LOX domain executing its catalytic activity[33]. However, recent studies identified several RNA-bound proteins lacking described RNA-binding domains[54,55], corroborating the idea that distinct RNA structures might be crucial for specific protein interactions. The latter point is of particular importance for lncRNAs, which are thought to adopt a variety of folds[56]. Exemplarily, the highly structured lncRNA SRA was shown to directly interact with estrogen receptors and a variety of other protein factors[57,58]. The need for higher order structures within functional RNAs is also reflected by our data, as the GATA6-AS-LOXL2 interaction was abolished upon denaturation or deletion of the sequence motives surrounding the identified binding sites.

Regarding the biological consequences resulting from the interplay between GATA6-AS and LOXL2, gene expression profiling showed ~71% of all genes deregulated by GATA6-AS silencing to be inversely regulated upon silencing of LOXL2. This implies a repressive function of GATA6-AS on LOXL2: In this scenario, silencing of GATA6-AS leads to increased LOXL2 deamination activity which consecutively results in reduction of H3K4me3 levels and transcriptional repression of these genes. The number of genes which were induced by GATA6-AS silencing but repressed by LOXL2-silencing was unambiguously lower and it remains to be determined whether these genes might be indirectly regulated or whether this regulation may be related to a repressive function of H3K4 methylation, as it occurs at some promoters in yeast[59,60]. Interestingly, we identified and validated a number of angiogenesis- and hypoxia-related genes to be downregulated by GATA6-AS silencing. Since the majority of these genes exhibits pronounced H3K4me3 peaks in their pro- moter region in HUVECs[61], this could provide a possible link between GATA6-AS-LOXL2-mediated gene regulation and our observed effects on endothelial cell functions. Indeed, we were able to show a significant decrease in H3K4me3 ChIP efficiencies upon GATA6-AS silencing at two of the identified angiogenesis- related genes that were inversely regulated. Moreover, hypoxia, in addition to inducing GATA6-AS expression to modulate nuclear LOXL2 function, was found to induce trimethylation of H3K4[62]. Even though, the question whether GATA6-AS is an allosteric inhibitor of LOXL2 or controls targeting of LOXL2 to the chro- matin needs further investigation, our biochemical data strongly supports a model in which the nuclear gene regulatory function of LOXL2 is negatively controlled by GATA6-AS (Fig. 6f).

Besides describing GATA6-AS mechanistically, we also uncovered interesting functional effects of this lncRNA in endothelial cells. Silencing of GATA6-AS reduced EndMT and augmented vessel formation in xenograft models in vivo. One may speculate that the regulation of EndMT may be a biologically conserved function of GATA6-AS since overexpression of this lncRNA was shown to inhibit tumor cell metastasis[63], a process involving the transition of epithelial cells to a mesenchymal state[64]. The inhibition of EndMT by GATA6-AS silencing may also explain our observed impairment of in vitro sprouting, which requires a transient induction of EndMT[65]. Nevertheless, since vessel sprouting is considered as a first initial stage of angio- genesis, the inhibition of sprouting and migration in vitro is counterintuitive to our in vivo findings, demonstrating increased vessel formation upon GATA6-AS silencing in endothelial cells. However, since the inhibition of sprouting is modest, the main- tenance of the endothelial integrity may have compensated for these negative effects of GATA6-AS silencing in vivo.

Taken together, we provide evidence that the lncRNA GATA6- AS is implicated in the epigenetic regulation of endothelial gene expression by binding to a fraction of nuclear LOXL2 and impairing its function as H3K4me3 deaminase, thus controlling endothelial cell function in vitro and in vivo.

## Methods

**Oligonucleotides**. The sequences of all oligonucleotides used in this study are given in Supplementary Table 5. Primers, DNA oligonucleotides and siRNAs were purchased from Sigma-Aldrich, LNA GapmeRs from Exiqon, and 2′O-Me-RNA probes from IDT.

**Cell culture**. Pooled HUVECs (Lonza) were cultured in endothelial basal medium containing EGM SingleQuots (Lonza) and 10% fetal calf serum (Invitrogen); HeLa cells in DMEM (Gibco) with 10% fetal calf serum and 1% penicillin/streptomycin. All cells tested negative for mycoplasma. Hypoxia was induced by incubation at 0.2% $O_2$ for 12 h or 24 h.

**Transfection and in vitro sprouting**. For silencing of gene expression, cells were transfected with LNA GapmeRs (50 nM; Exiqon) or siRNAs (67 nM; Sigma-Aldrich) using Lipofectamine RNAiMax (Life Technologies) according to the manufacturer's protocol. Endothelial angiogenesis was studied by in vitro spheroid sprouting assays as described before[66]. Briefly, HUVECs were transfected with GapmeRs or siRNAs for 24 h and cells were trypsinized, added to a mixture of culture medium and methylcellulose (80%:20%) and transferred to a 96-well plate to allow for the for- mation of spheroids. After 24 h at 37 °C, spheroids were collected, added to methylcellulose supplemented with FCS (80%:20%) and embedded in a collagen type-I gel (BD Biosciences). After incubation for 24 h at 37 °C under basal conditions or VEGFA stimulation (50 ng/ml), gels were fixed with 10% formaldehyde in PBS and images were taken, using an Axio Observer Z1.0 microscope (Zeiss) at ×10 magnification. Cumulative sprout length of each spheroid was measured as read out for in vitro angiogenesis by using AxioVision Version 4.6 (Carl Zeiss) digital imaging software. 10 spheroids were analyzed per group and experiment.

**BrdU cell proliferation assay**. Cell proliferation assays were performed using the BrdU Flow Kit (BD Pharmingen) according to manufacturer's instructions. Briefly, transfected HUVECs were washed with PermWash buffer and incubated with BrdU (0.1 mM) for 3 h at 37 °C. Next, cells were washed thoroughly with PBS, Cytofix/Cytoperm buffer, PermWash buffer, and CytopermPlus buffer. Subse- quently, cells were incubated in DNase I solution for 1 h at 37 °C, washed with PermWash buffer and further incubated with BrdU-V450 for 20 min at RT. Finally, 7-AAd was added for 10 min at RT and cells were analyzed, using a FACS Canto II device (BD Bioscience).

**Caspase 3/7 assay**. Caspase 3/7 activity in HUVEC cell culture was measured using Caspase-Glo 3/7 assays (Promega) according to the manufacturer's protocol. Transfected HUVECs were incubated with Caspase 3/7 reagent diluted 1:100 in reaction buffer and incubated for 1 h at 37 °C. Measurements were conducted with a GloMax-Multi Microplate Multimode Reader.

**Lentiviral overexpression of GATA6-AS**. For lentiviral overexpression, GATA6- AS (GRCh37; NR_102763.1, for sequence, Supplementary Table 6) was first cloned into pLenti4/V5-DEST (GeneArt™) and subsequently, lentiviral particles were generated by transfection of HEK293T cells with p8.91, pMD2, and pLenti4/V5- DEST-GATA6-AS plasmids following incubation for 4 days at 37 °C. Empty pLENTI/V5-DEST was used as control. Supernatants were concentrated by ultracentrifugation (Optima™ XPN; Beckman Coulter) for 140 min at 20,500 × g at 4 °C, using a SW 32 Ti rotor.

**RNA antisense affinity selection and mass spectrometry**. HeLa cells were lysed in lysis buffer (50 mM Tris-HCl pH8, 50 mM NaCl, 0.5% NP-40, 80U Ribolock, protease inhibitor) and volumes were adjusted to 1 ml with the same buffer lacking NP-40. For selection of RNP complexes, lysates were pre-cleared for 2 h at 4 °C and subsequently incubated with 100 pmol 2′O-Me-RNA oligonucleotides for 1 h at 37 °C. RNP-oligonucleotide complexes were captured using 25 μl pre-blocked (yeast tRNA, glycogen; both 0.2 mg/ml) streptavidin C1 beads (Thermo Fisher) for 1 h at 37 °C. Beads were washed thoroughly with washing buffer (50 mM Tris-HCl pH8, 50 mM NaCl, 0.05% NP-40) and biotin (50 μM) eluted at RT. Eluates were analyzed by RT-qPCR and mass spectrometry using a high resolution quadrupole Orbitrap mass spectrometer[67] (Q Exactive, Thermo Fisher).

**RNA immunoprecipitation**. HUVECs were washed with PBS, $UV_{254}$-irradiated (2 × 50 mJ/cm$^2$; Stratalinker 2400, Stratagene) and lysed (50 mM Tris-HCl pH8, 50 mM NaCl, 0.5% NP-40, 80U Ribolock, protease inhibitor) for 30 min on ice. Supernatants were cleared for 5 min at 20,000 × g and adjusted to 1 ml with the same buffer lacking NP-40. For immunoprecipitation, 30 μl protein G Dynabeads

(Thermo Fisher) were first coupled with 15 µg LOXL2 or serotype control antibodies (AF#2639, AB-108-C; R&D) and subsequently incubated with lysates for 4 h at 4 °C. Beads were washed three times with washing buffer (50 mM Tris-HCl pH8, 50 mM NaCl, 0.05% NP-40) and RNA was recovered by proteinase K digestion (30 min, 55 °C), phenolization and ethanol precipitation.

**RNA isolation and RT-qPCR**. Total RNA from cells was isolated and DNase digested using miRNeasy kits (Qiagen). Reverse transcription was carried out using 500 ng RNA, random hexamers and MuLV reverse transcriptase (Thermo Fisher). Subsequent Fast SYBR Green qPCRs were performed on StepOnePlus real-time PCR systems (Thermo Fisher). RPLP0 amplification was used for data normalization. Relative expression levels were calculated by $2^{-\Delta Ct}$.

**Chromatin immunoprecipitation**. HUVECs were formaldehyde crosslinked at RT (1% in PBS) and reactions were quenched with glycine after 10 min. Next, cells were lysed in cytoplasmic lysis buffer (50 mM HEPES pH 7.4, 140 mM NaCl, 1 mM EDTA, 0.5% NP-40, 10% glycerol, 0.25% TritonX-100, protease inhibitor) and nuclei were pelleted for 10 min at $1000 \times g$ and lysed in nucleic lysis buffer (10 mM Tris-HCl pH 7.6, 1 mM EDTA, 0.1% SDS). Nuclear extracts were sonified (duty cycle 2%, 105 W, cycles/burst: 200; Covaris S220) and cell debris was pelleted at $20,000 \times g$ for 10 min. Recovered supernatants were diluted with dilution buffer (20 mM HEPES pH 7.4, 1 mM EDTA, 150 mM NaCl, 1% TritonX-100, 0.1% SDS) and incubated with 2 µg antibodies (#8580; Abcam, or #12-370; Millipore). Immunoprecipitation was carried out using pre-blocked protein A/G agarose beads (Diagenode) for 2 h at 4 °C. Finally, immunocomplexes were washed with high salt buffer (20 mM HEPES pH8, 1 mM EDTA, 150 mM NaCl, 1% TritonX-100, 0.1% DOC), low salt buffer (20 mM HEPES pH8, 1 mM EDTA, 500 mM NaCl, 0.1% SDS, 1% TritonX-100, 0.1% DOC), LiCl buffer (20 mM HEPES pH8, 250 mM LiCl, 0.5% NP40, 0.5% DOC), and washing buffer (20 mM HEPES pH8, 1 mM EDTA). Crosslinking was reversed by proteinase K digestion (2 h, 55 °C) and DNA was purified using MiniElute Reaction Cleanup Kits (Qiagen).

**Deep sequencing**. For RNA deep sequencing, normoxic and hypoxic HUVEC ribosomal RNA-depleted RNA was fragmented and primed with random hexamers[42]. The sequence library was constructed by utilizing the Illumina TruSeq Stranded Total RNA Library Prep Kit, according to the manufacturer's protocol. Reads were mapped with Tophat2 version 2.0.6 to GRCh37. Gene expression estimation was performed by Cufflinks version 2.1 with default parameters. Data was filtered for long non-coding RNAs, a significant hypoxic response, and FPKM values greater 0.1.

**Endothelial cell migration assays**. Transwell-inserts with a pore size of 8 µm (Greiner Bio-One) were coated with 0.2% gelatin for 1 h at 37 °C and washed with PBS. HUVECs were trypsinized and 50,000 cells were added to the inserts and allowed to migrate to the lower part of the chamber for 3 h at 37 °C in medium containing FCS and VEGFA (50 ng/ml) in the lower chamber. Cells on the topside of the inserts were removed and migrated cells were fixed with methanol for 20 min at RT and stained with Hoechst. The number of migrated cells was captured by fluorescence microscopy, counting five individual fields of view per transwell.

**Exon arrays and bioinformatics**. Gene expression was analyzed using GeneChip Human Exon 1.0 ST arrays (Affymetrix). In short, RNA was isolated from HUVECs transfected with GapmeRs or siRNAs for 48 h and samples were processed according to the manufacturer's protocol. CEL files were uploaded to and analyzed by the non-coder web interface (http://noncoder.mpi-bn.mpg.de)[68] with the default parameters.

**Cellular fractionation**. HUVECs were fractionated as described elsewhere[69]. Briefly, cells were washed twice with PBS and centrifuged for 5 min at $500 \times g$. Pellets were resuspended in cytoplasmic lysis buffer (10 mM Tris-HCl pH8, 10 mM NaCl, 1.5 mM MgCl$_2$) and incubated on ice for 5 min. Following centrifugation for 5 min at $1000 \times g$, cytoplasmic supernatants were taken. Nuclear pellets were washed two times with PBS and resuspended in nucleic lysis buffer (50 mM Tris-HCl pH8, 50 mM NaCl, 0.5% NP-40, protease inhibitor) and kept on ice for additional 5 min. Following centrifugation for 5 min at $20,000 \times g$ nucleic supernatant was taken and RNA was isolated from both fractions.

**RNA accessibility assays**. HUVECs were lysed in lysis buffer (50 mM Tris-HCl pH8, 150 mM NaCl, 0.5 % NP-40, protease inhibitors) and subsequently adjusted to a total volume of 1 ml with final concentrations of 60 mM NaCl, 50 mM Tris-HCl pH8, 75 mM KCl, 3 mM MgCl$_2$, 10 mM DTT and protease inhibitor. Next, 100 µl of the adjusted lysate were mixed with 100 pmol DNA oligonucleotides and incubated for 2 h at 4 °C. Thereafter, 2.5 U RNase H (NEB) were added and reactions were kept for 20 min at 37 °C. Finally, RNA was isolated for RT-PCR.

**Individual nucleotide resolution crosslinking and immunoprecipitation**. The iCLIP assay was carried out as described elsewhere[70]. Briefly, a 15 cm dish of HUVECs was washed with PBS and UV-irradiated ($1 \times 150$ mJ/cm$^2$ at 254 nm)

using a Stratalinker 2400 device (Stratagene). Next, nuclear extract was prepared and subjected to DNase and RNase treatment using 75 U of RNase I (Ambion). For RIP, 4 µg LOXL2 or serotype control antibodies (AF#2639, AB-108-C; R&D) were coupled to 30 µl of Protein G Dynabeads (Thermo Fisher) and incubated with nuclear extract for 2 h at 4 °C. After washing four times with washing buffer (50 mM Tris-HCl pH7.4, 1000 mM NaCl, 0.05% Tween 20), the co-immunoprecipitated RNA was dephosphorylated, ligated with a 3′-linker and 5′-radiolabeled with [γ-$^{32}$P]-ATP. Samples were size-separated by neutral SDS-PAGE (NuPAGE, Invitrogen) and transferred to a nitrocellulose membrane. Protein-RNA complexes were visualized by autoradiography. LOXL2-RNA complexes were cut from the membrane, proteins were digested with proteinase K and RNA was subjected to iCLIP library preparation as described elsewhere[70]. A 150 M library was sequenced on an Illumina Nextseq500 using Mid Flowcell V2 with $1 \times 150$ bp read length (single read). Reads were mapped to the GRCh38 transcriptome by Bowtie2 (version 2.2.6) followed by duplicate deletion with samtools (version 1.3).

**Western blot**. Whole cell lysates were prepared using lysis buffer (50 mM Tris-HCl pH8, 50 mM NaCl, 0.5% NP-40, protease inhibitor) and protein concentrations were determined by Bradford assays. Proteins were separated by SDS-PAGE and transferred to PVDF membranes. Antibodies detecting LOXL2 (AF#2639; R&D; 1:200), H3 (#1791; Abcam; 1:2000), H3K4me3 (#8580; Abcam; 1:1000), GATA6 (D61E4; Cell Signaling; 1:300), pAkt (#9271 S; Cell Signaling; 1:2000), Akt (#9272; Cell Signaling; 1:2000), β-Actin (#3280; Abcam; 1:2000), or Tubulin (RB-9281-P1; Thermo Fisher; 1:1000) were incubated overnight at the indicated dilutions. For detection, HRP-coupled secondary antibodies were used. Extracellular LOXL2 was measured in supernatants from HUVECs transfected with GapmeRs as described before. 24 h after transfection, medium was replaced with EBM (0.1% FCS) and cells were incubated for another 24 h. Supernatants were collected and cleared for 5 min at $4000 \times g$. Finally, extracellular proteins were concentrated by ultracentrifugation (Optima MAX-XP; Beckman Coulter) for 150 min at $200,000 \times g$ at 4 °C, using a TLA-110 rotor and analyzed by western blotting. Original immunoblots are depicted in Supplementary Figs. 8, 9.

**In vitro transcription and His-tag pulldown**. Full length GATA6-AS (1788nt) or a truncated transcript (735nt) lacking nucleotides 1–642 and 1378–1788 (for sequences, see Supplementary Table 6) were cloned into pCR2.1 TOPO for run-off in vitro transcription using the RiboMAX Large Scale RNA Production System-T7 (Promega) according to the manufacturer's protocol. Transcripts were gel purified and folded (5 min 65 °C → 10 min RT → 30 min 30 °C → 15 min 37 °C) in interaction buffer (20 mM Tris-HCl pH8, 5 mM MgCl$_2$, 1 mM DTT, 0.01% NP-40, 48.75 mM NaCl, protease inhibitor) or denatured (2 min 95 °C → ice) previous to His-LOXL2 pulldown. For His-tag pulldown, 10 pmol recombinant His-LOXL2 (2639-AO-010; R&D) were incubated with 1 pmol of transcript for 30 min at 30 °C and subsequently for 15 min at 37 °C in interaction buffer. Next, reactions were adjusted to 150 mM NaCl and incubated with 25 µl (pbv) equilibrated Ni-NTA agarose beads (Qiagen) for 3 h at 4 °C. After washing with washing buffer (20 mM Tris-HCl pH8, 5 mM MgCl$_2$, 1 mM DTT, 0.01% NP-40, 150 mM NaCl) beads were eluted in 200 µl 1× proteinase K buffer, RNA was recovered by phenol-chloroform extraction and ethanol precipitation and used for RT-qPCR or Northern blotting.

**Northern blot**. For Northern blotting, RNA was separated on a 5% denaturing polyacrylamide gel, electroblotted to a nylon membrane (Roth), UV-crosslinked at 254 nm and pre-incubated in church buffer for 1 h at 50 °C. DIG-labeled dsDNA probes were hybridized overnight at 50 °C. After extensive washing steps (2× SSC, 0.1% SDS and 0.5× SSC, 0.1% SDS), RNA was visualized using AP-coupled anti-DIG Fab fragments (1:10,000; Roche) and 1% CSPD detection solution.

**Statistical analysis**. Data are expressed as mean ± S.E.M. and statistical significance was assessed by two-tailed Student's t test and Mann–Whitney U test. Probability values of less than 0.05 were considered significant. n refers to the number of replicates.

**Endothelial-mesenchymal transition**. $4 \times 10^5$ HUVECs were seeded in 60 mm cell culture dishes and transfected with 50 nM LNA GapmeRs. After 24 h, cells were transferred to SingleQuots-supplemented EBM (Lonza) or differentiation medium (supplemented EBM containing 10 ng/ml TGF-β2 and lacking BBE and EGF). After 72 h of incubation RNA was isolated for RT-qPCR.

**Immunofluorescence**. $1.53 \times 10^4$ HUVECs were seeded into 8-chamber glass slides, coated with human fibronectin (1 µg/ml; Sigma-Aldrich). 24 h after seeding, HUVECs were transfected with 50 nM LNA GapmeRs and endothelial-to-mesenchymal transition was induced as described above. Finally, cells were fixed using 4% PFA and permeabilized with 0.1% TritonX-100. After blocking with 10% donkey serum, primary antibodies targeting VE-cadherin (1:200; #2500 S; Cell signaling) or SM22 (1:100; #10135; Abcam) were incubated overnight at 4 °C. Secondary antibodies were incubated for 1 h at RT and nuclei were counter stained using Hoechst diluted in Fluoromount (Sigma-Aldrich). Imaging was done using an Observer.Z1 microscope (Carl Zeiss). Analysis was performed by the measurement VE-cadherin intensity using ImageJ.

**Animal experiments**. All mice experiments were carried out in accordance with the principles of laboratory animal care as well as according to the German national laws. The studies have been approved by the local ethic committee (Regierungspräsidium Darmstadt, Hessen). The hind limb ischemia model was performed as described elsewhere[71], using 10 to 12 weeks old C57BL/6 J mice. Briefly, the proximal femoral artery as well as the deep branch and the distal saphenous artery were ligated. After 21 days, mice were sacrificed and leg muscle tissue was used for RNA isolation. Swiss mouse embryos were taken out at E12.5 and five hearts were pooled for RNA isolation.

**Matrigel plug assay**. Matrigel plug assays were performed as described elsewhere[27]. Briefly, 8-week-old to 10-week-old NMRI nu/nu mice were injected subcutaneously with two matrigel basement matrix plugs containing spheroids derived from GATA6-AS-silenced or LNA control HUVECs. Matrigel plugs were harvested at day 21. In order to analyze perfused capillaries and stain for mouse endothelial cells, biotinylated isolectin B4 (200 µg/mouse) was injected 15 min before harvest. Subsequently, 4 µm paraffin sections were stained for human endothelial cell marker (ULEX), mouse endothelial marker isolectin-B4 and cell nuclei (Hoechst). Ulex-positive and isolectin B4-positive structures were counted manually in five microscopic fields in one section of each plug. Images were generated using a laser scanning confocal microscope (LSM 780, Carl Zeiss) at a magnification of 20×. The average vessel number per plug was calculated as follows; Ulex-positive or isolectin B4-positive structures, both sharing a positive nuclear Hoechst signal were counted and the mean value for both matrigel plugs for each mouse was taken for the statistical analysis using a Mann-Whitney U test.

**Data availability**. The Exon array data and RNA deep sequencing data that support the findings of this study have been deposited in the Gene Expression Omnibus database with the identifier "GSE107033". The LOXL2-iCLIP dataset and the rest of the data are available from the corresponding author upon request.

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

## Acknowledgements

We thank Ariane Fischer, Marion Muhly-Reinholz and Soraya Hölper for technical support and Michaela Müller-McNicoll for conceptual help. This work was supported by the SFB902 (DFG), the ERC advanced grant Angiolncs, the LOEWE program Medical RNomics (State of Hessen) to S.D., O.R. and A.B. as well as the Excellence Cluster Cardiopulmonary Systems (DFG) to S.D.

## Author contributions

P.N. and N.J. designed and performed experiments, analyzed data, and drafted the article. A.K., S.F.G., Y.F. and O.R. performed experiments. D.J. analyzed data. R.A.B., M.K., A.B. and P.G. revised the article critically. S.D. oversaw the project, designed experiments, analyzed data and drafted the article.

## Additional information

**Competing interests:** N.J. and S.D. applied for a patent on the use of long non-coding RNAs for the treatment of endothelial dysfunction. The remaining authors declare no competing financial interests.

