## [Peer Review File · Nature Communications]

Reviewers' comments:

Reviewer #1 (Remarks to the Author):

The authors have presented a paper that identifies the transcript GATA6-AS1 as regulated in a screen for endothelial cell transcripts activated in hypoxia. They show that this transcript does not regulate GATA6 as one might expect. They used a range of experiments to show LOXL2 as an interacting protein, confirmed the interaction in HeLa cells and EC and then went on to show that this function was via the nuclear mechanism of LOXL2 and not the cytoplasmic/extracellular mechanism. The effect was to modulate epigenetic activation of genes (2 confirmed) in EC linking to the sprouting angiogenesis phenotype.

There are some novel aspects to the work - the identification of GATA6-AS1, the mechanism of binding to LOXL2 and the mechanism of regulation of angiogenesis-associated genes. However, there are also some caveats and limitations that require further study.

1. All of the endothelial work appears to be in HUVEC. At least parts need to be performed in another EC primary cell of more relevance to human angiogenesis.
2. As shown in the Figures, there are a number of isoforms of GATA6-AS in the locus. One does not get from the data presented, the biological understanding of the expression levels of each isoform under basal and stimulated conditions. Although this is often inaccurate from the annotations, biological experimentation is required to firm up expression of each isoform in EC. Both level and which isoform. This is important as one needs to appreciate and understand the axis regulation and how this impacts on mechanism.
3. How does the hypoxia regulation link back to expression increase in GATA6-AS1? Where are the sites and transcriptional control that can explain the regulation? At least informatics approaches needed and visualisation in the Figures.
4. For the mass spec, the number of unique peptides ascertained for each protein should be shown.
5. It is a little concerning that some of the data presented has been assessed statistically, but is not actually significant but has been interpreted on having a difference. e.g. Fig 5a and e. This really links to the reader that the data becomes slightly weaker as one progresses the paper. Although the work is novel, the data appears to weaken. Perhaps there is a broader relevance to angiogenic genes and actually just assessing 2 of them experimentally is simply diluting the importance.
6. For the hypoxia regulation shown in Figure 1b, a longer time course should add value.
7. The images of sprouting angiogenesis look quite marginal. Where were these experiments blinded?
8. The experimental evidence of effects on angiogenesis and activation tags would be reassuring with an overexpression strategy, under basal and stimulated conditions.
9. The GAPMERS, how do they affect the different transcripts of the GATA6-AS1 transcript locus?
10. The axis looks as though it is conserved in mouse. Although this requires some thought and validation, the authors should consider whether this is A. relevant and correct and B. A conserved function of the lncRNA.

Reviewer #2 (Remarks to the Author):

The manuscript is of interest and well describes the complexity of the mechanism of action of a lncRNA identified in hypoxic conditions, which influences more than one biological process at various regulation levels.

Criticisms:

1. RNA seq was conducted in HUVEC cells in hypoxic conditions; RIP experiments were conducted in HUVEC in normoxic conditions; CLIP and pull-down experiments in HeLa cells in normoxic conditions. While all these experiments prove that the lncRNA can bind loxl2 in all these instances, it does not prove its specific action in hypoxic conditions in HUVEC. Thus, the authors should perform a RIP for loxl2 in HUVEC cells in hypoxic conditions and compare the enrichment for the lncRNA between normoxic and hypoxic conditions. If binding is specific of hypoxic conditions, there should be an increased binding between the epigenetic protein and the lncRNA.
2. The authors silence both the lncRNA and loxl2 and show with RNAseq and ChIP seq for H3K4me3 that there is a negative correlation between the expression of loxl2 target genes and the silencing of the two genes. The specificity of the data could be improved by overexpressing the lncRNA in HUVEC in normoxia, comparing then RNAseq and ChIPseq results with those obtained treating cells with loxl2 siRNA. Are the genes modified by loxl2 silencing direct targets of this epigenetic factors? This needs to be clarified.
3. Most of the study is in vitro; the value of the manuscript would increase significantly if the authors could confirm some of the data also in vivo, by RIP for instance, in models of ischemia.

Minor:

1. Fig. 1 shows that the silencing of the lncRNA causes a significant reduction of angiogenic sprouting in the normoxic condition, suggesting a role of this lncRNA in angiogenesis that is not limited to hypoxia. Can the author speculate on this aspect? Is the effect increased in hypoxic condition?
2. Can the author confirm the nuclear localization of this lncRNA with another technique? For instance, FISH.
3. The authors state that this lncRNA does not work in "cis". Have they tried to determine this in hypoxic conditions? Please explain better in the text.

Reviewer #3 (Remarks to the Author):

The manuscript from Neumann et al show how the lncRNA GATA6-AS interacts with LOXL2 acting as a negative regulator of LOXL2 activity in endothelial cells under hypoxia conditions. Through this interaction the lncRNA GATA6-AS is able to regulate endothelial gene expression. Although the results are very interesting, the model proposed has not been completely demonstrated by the current evidences and further experiments are required.

Major concerns

The authors showed how under hypoxia there is an up-regulation of several lncRNA and they focus their attention to the lncRNA GATA6-AS.

Figure 1d-f: All the experiments shown in these panels have to be done under hypoxia conditions, the situation in which the authors claim that the lncRNA GATA6-AS is induced and control endothelial gene expression.

Comment: This situation happens throughout the manuscript, most of the experiments shown have been done under normoxia conditions. For the reviewer is then not clear if the lncRNA GATA6-AS is also expressed under normoxia and if so is its function in normoxia different than

under hypoxia?

Gene Ontology analysis from the experiment showed in figure S4C can be indicated here. GO terms have to be related with the pro-angiogenic function suggested for the lncRNA GATA6-AS.

Figure S2: Here the authors suggest that lncRNA GATA6-AS is not acting in cis, however again this experiment has to be done under hypoxia conditions

Figure 3: The authors demonstrate the interaction between lncRNA GATA6-AS and LOXL2 in HeLa lysates and in HUVEC. In panel a the authors should show LOXL2 levels (input) in the control (scramble) and GATA6-AS conditions. In panel b the authors should show by western-blot His-LOXL2 expression.

Figure 4. It is known the LOXL2 affects cellular proliferation and cell death, the effect showed by the authors in spheroid sprouting assays can be related with cell proliferation rather than angiogenesis. Moreover, based on gene expression profile if LOXL2 and lncRNA GATA6-AS share sets of inversely regulated genes, and lncRNA GATA6-AS is pro-angiogenic, LOXL2 should be anti-angiogenic. If this is the case, why the absence of LOXL2 results in a sprouting phenotype? This is the same phenotype shown when lncRNA GATA6-AS is knocked down.

Figure 5. In this figure the authors try to demonstrate how the interaction between LOXL2 and lncRNA GATA6-AS interferes with LOXL2 activity.

Fig5a: P value? Is this decrease significant? Again, this reduction in the global levels of H3K4me3 have to be shown under hypoxia. Having say that, sometimes is difficult to observe changes in histone marks at global levels, probably a limited number of genes will be under the control of LOXL2- GATA6-AS.

Loading control must be total histone H3 instead of tubulin.

Fig5e: P-value? For the reviewer is not clear if this decrease is significant. Is the ChIP-PCR corrected by the total amount of H3? Irrelevant genomic regions have to be shown. Also it would be nice to show a control positive region, a LOXL2 target promoter (End-Cadherin maybe?). LOXL2 ChIP in these suggested regions is also a must. Again, the effects should be increase under hypoxia conditions.

To demonstrate that LOXL2 activity decreases in vitro assays have to be done. The authors can use the Amplex-Red kit that measures the levels of H₂O₂ released upon LOXL2 reaction. Recombinant LOXL2 protein + lncRNA GATA6-AS + substrate (H3K4me3 peptides) can be used. Also by western blot the authors can check in nucleosomes incubated with LOXL2 in the presence or absence of lncRNA GATA6-AS histone marks levels.

Minor concern

Herranz et al, Febs J must be cited, this is the paper that describes the role of LOXL2 as a histone deaminase protein.

Reviewer #4 (Remarks to the Author):

In this study, the authors identified 64 hypoxia-responsive long non-coding RNAs (lncRNAs) in HUVEC cells. Among them, GATA6-AS is an uncharacterized transcript that showed consistent response (up-regulation) to hypoxia treatment for either 12h or 24h. The authors go on further to characterize the biological function of GATA6-AS and demonstrate that GATA6-AS localized mainly

in the nuclear and played pro-angiogenic functions in HUVEC cells. By performing GATA6-AS affinity selection and mass spectrometry in HeLa cells, the authors identified multiple protein binding partners of GATA6-AS in which Lysyl oxidase-like 2 (LOXL2) was among the top enriched proteins. The authors then performed GATA6-AS affinity selection followed by western blotting, RNA immunoprecipitation (RIP), in vitro His-tag pulldown assay followed by northern blotting and individual-nucleotide resolution crosslinking and immunoprecipitation (iCLIP) to confirm the interaction between GATA6-AS and LOXL2. Two LOXL2 iCLIP-tags were identified on GATA6-AS. It appears that GATA6-AS silencing did not affect intra- or extracellular LOXL2 levels although there's a direct binding between them. Given LOXL2 has been reported to play nuclear function as a transcription co-repressor through deaminating H3K4me3, the authors examined the expression of H3K4me3 in HUVECs and found modest reduction (17%) of H3K4me3 after GATA6-AS silencing. The authors then performed exon arrays to study endothelial gene expression profile in either GATA6-AS- or LOXL2-silenced cells. And they found an inversely regulated gene expression pattern between GATA6-AS- or LOXL2-silenced groups. Finally, the authors picked one from each group (COX-2 and POSTN) for H3K4me3 ChIP-PCR analysis and found GATA6-AS silencing reduced H3K4me3 ChIP efficiencies on the promoter regions of COX-2 and POSTN. Taken together, the authors drove the conclusion that GATA6-AS is a hypoxia-responsive transcript, and that GATA6-AS epigenetically regulates endothelial gene expression and plays pro-angiogenic roles through interacting with LOXL2.

Overall, this is a study consisting of extensive amount of work. However the possible mechanism has modest novelty since LOXL2 has been well characterized in hypoxic response and angiogenic function in endothelial cells, so is LOXL2's transcriptional repressing function through H3K4me3. In addition, several observations presented here are not convincing enough to support the conclusions. There are a number of points that need to be addressed as outlined below.

1. The subcellular localization of GATA6-AS in HUVEC cells was determined by cellular fractionation in Fig. 1D. It would be more convincing if the nuclear localization of GATA6-AS can be confirmed by RNA-FISH in HUVECs under both normoxic and hypoxic condition.
2. The authors studied the biological functions of hypoxia-responsive GATA6-AS in HUVEC cells. It would be nice to see if GATA6-AS also plays pro-angiogenic effect in in vivo settings, such as hindlimb ischemia model, retinal hypoxia, etc. In vivo studies would have more clinical impact for the current study.
3. GATA6-AS was identified as hypoxia-responsive transcript in endothelial cell, also the main conclusion of the current study is that GATA6-AR regulates endothelial gene expression. However, the authors switched to HeLa cells for the antisense affinity selection when trying to identify protein binding partners of GATA-AR (Fig. 2d) and repeated the affinity selection of GATA6-AS also in HeLa cell lysates (Fig. 3a). For the following RIP and iCLIP assays, the authors switched back to HUVECs. It is a somewhat strange experiment design. It would be more reasonable if the affinity selection was conducted using HUVECs.
4. The authors employed four independent assays to confirm the interaction between GATA6-AS and LOXL2, which is very impressive. However it would be more convincing if the nuclear localization of LOXL2 can be confirmed in HUVEC cells under normal and hypoxic condition, given it is a frequently raised question that how nuclear localized lncRNAs bind cytosolic localized RNA/proteins. Also it would be interesting to know whether the interaction between GATA6-AS and LOXL2 would have any difference under physiological and pathological conditions.
5. Fig. 4C needs better western blot image with quantification results. Tubulin looks uneven in the current blot.
6. I wonder whether a p value has been calculated for H3K4me3 quantification result presented in Fig. 5a. It appears that the differences between LNA Ctl and LNA GATA6-AS groups are not significant.
7. The ChIP-PCR analysis results (Fig. 5e) are not convincing either (no P value?). Also sample size (n=3) is highly recommended for gene expression profiling studies (Fig. S4c).

RESPONSE TO THE REVIEWERS

Response to Reviewer #1:

The authors have presented a paper that identifies the transcript GATA6-AS1 as regulated in a screen for endothelial cell transcripts activated in hypoxia. They show that this transcript does not regulate GATA6 as one might expect. They used a range of experiments to show LOXL2 as an interacting protein, confirmed the interaction in HeLa cells and EC and then went on to show that this function was via the nuclear mechanism of LOXL2 and not the cytoplasmic/extracellular mechanism. The effect was to modulate epigenetic activation of genes (2 confirmed) in EC linking to the sprouting angiogenesis phenotype.

There are some novel aspects to the work - the identification of GATA6-AS1, the mechanism of binding to LOXL2 and the mechanism of regulation of angiogenesis-associated genes. However, there are also some caveats and limitations that require further study.

1. All of the endothelial work appears to be in HUVEC. At least parts need to be performed in another EC primary cell of more relevance to human angiogenesis.

Answer: According to the reviewer's request, we analyzed the regulation of GATA6-AS in human cardiac microvascular endothelial cells (HCMECs) after 12h and 24h of hypoxia (0.2% O₂) and found GATA6-AS to be induced in both settings, compared to normoxic control conditions. This new data is shown in Fig. 1a for the Reviewer. For additional functional assays, HCMECs were used in transwell migration studies showing that silencing of GATA6-AS regulates the migratory capacity also in human cardiac microvascular endothelial cells (Fig. 1b, c for the Reviewer).

2. As shown in the Figures, there are a number of isoforms of GATA6-AS in the locus. One does not get from the data presented, the biological understanding of the expression levels of each isoform under basal and stimulated conditions. Although this is often inaccurate from the annotations, biological experimentation is required to firm up expression of each isoform in EC. Both level and which isoform. This is important as one needs to appreciate and understand the axis regulation and how this impacts on mechanism.

Answer: To address the important question whether hypoxia selectively affects individual GATA6-AS transcript variants, we first designed primers covering the GATA6-AS locus which can be used in competitive RT-PCR (Fig. 2a for the Reviewer). By comparing normoxic versus hypoxic (0.2% O₂, 24h) conditions, we demonstrate that hypoxia induces the transcript variant GATA6-AS-001 (Fig. 2b, c for the Reviewer) as well as alternatively spliced isoforms, which comprise exon 2' and exon 5' of transcript variant GATA6-AS-002, but also a spliced transcript comprising exon 2' and exon 5 (which is so far not annotated) (Fig. 2d, e for the Reviewer). Of note, all of these transcripts were silenced by GapmeR treatment (Fig. 2b-e for the Reviewer).

Second, we characterized the transcripts at the GATA6-AS locus by analyzing RNA sequencing data. Data evaluation revealed that the GATA6-AS-001 transcript surely is expressed in endothelial cells (Fig. 2f for the Reviewer). This was also confirmed by RT-PCR targeting the first and second exon of this transcript (Fig. 2b, c for the Reviewer). The identification of the transcript variants 002, 003, and 004 turned out to be more complex and likely the annotation of these transcripts may not be

complete. Using forward primers directed against the 5' parts of exons 2, 2' and 2'' together with reverse primers targeting exons 5 and 5' demonstrated transcript variants which must contain exon 2' or 2'', at least part of exon 3' (since they are silenced by GapmeRs), and exon 5 or 5' (see Fig. 2d, e for the Reviewer). Finally, visualization of the reads obtained by RNA sequencing clearly demonstrates that exon 1, exon 2'', exon 3, exon 3', exon 5, and exon 5' are all detected and are up-regulated by hypoxia (Fig. 2f for the Reviewer).

Additionally we quantified isoform expression using a serial dilution of in vitro transcribed full length GATA6-AS. As revealed by RT-PCR, the region of GATA6-AS which is covered by primer pair F1-R1 was found at a concentration of about 0.6pg/ μ g HUVEC total RNA, which equals approximately 2 to 6 copies per cell under normoxic conditions.

3. How does the hypoxia regulation link back to expression increase in GATA6-AS1? Where are the sites and transcriptional control that can explain the regulation? At least informatics approaches needed and visualisation in the Figures.

Answer: *In silico* prediction of transcription factor binding sites upstream of GATA6-AS suggest that various factors could interact with this region, including the hypoxia-inducible factor HIF1 α , which is well-known to regulate multiple genes (see Supplementary Fig. 1d). Therefore, we first used siRNAs targeting HIF1 α and analyzed expression of GATA6-AS under hypoxic conditions by RT-qPCR. Silencing of this transcription factor did not interfere with GATA6-AS expression, as shown in Supplementary Fig. 1e, suggesting that the hypoxia-induced up-regulation of GATA6-AS is independent of HIF1 α . In a next step, we analyzed the GATA6-AS promoter region in publicly available CHIP datasets from HUVECs and refined our list of possible transcription factors (see Fig. 3 for the Reviewer). Among these factors, we found candidates, which are known to be implicated in PI3K/Akt signaling (e.g. E2F1 or EGR1). Treatment of HUVECs with the Akt inhibitor MK-2206, which inhibits auto-phosphorylation and Akt-mediated phosphorylation of downstream signaling molecules, reduced hypoxia-driven GATA6-AS induction, suggesting that GATA6-AS expression is primarily controlled by HIF1 α -independent Akt signaling. These new experimental findings and a visualization of the identified transcription factor binding sites are shown in revised Fig. 1e and Supplementary Fig. 1d, e, f.

4. For the mass spec, the number of unique peptides ascertained for each protein should be shown.

Answer: *We thank the reviewer for this important remark and added the unique peptide scores from our mass spectrometry results to Table S3 of the revised manuscript.*

5. It is a little concerning that some of the data presented has been assessed statistically, but is not actually significant but has been interpreted on having a difference. e.g. Fig 5a and e. This really links to the reader that the data becomes slightly weaker as one progresses the paper. Although the work is novel, the data appears to weaken. Perhaps there is a broader relevance to angiogenic genes and actually just assessing 2 of them experimentally is simply diluting the importance.

Answer: We understand the concerns of the reviewer and increased the number of replicates for the analysis of global H3K4me3 levels to n=6. As shown in Fig. 6a of the revised manuscript, the observed reduction of H3K4me3 levels upon silencing of GATA6-AS is now significant. In addition, we increased the number of replicates for the H3K4me3 ChIP experiments to n=10 and observed a statistically significant reduction in the relative enrichment of PTGS2 and POSTN promoter regions upon silencing of GATA6-AS (see Fig. 6e of the revised manuscript). Finally, we analyzed the promoter regions of cysteinyl leukotriene receptor 2 (CYSLTR2), vascular cell adhesion molecule 1 (VCAM1) and plasminogen activator, tissue type (PLAT), all of which are implicated in regulation of endothelial cell functions and were among the GATA6-AS-LOXL2 inversely regulated gene set. As shown in Fig. 4 for the Reviewer, the promoter regions of these genes have pronounced H3K4me3 marks in HUVECs which matches with our suggested model of GATA6-AS-LOXL2-mediated gene regulation.

6. For the hypoxia regulation shown in Figure 1b, a longer time course should add value.

Answer: We thank the reviewer for this comment and performed a hypoxia time course experiment covering 24h, 48h, and 72h of hypoxia. As seen by RT-qPCR, GATA6-AS is significantly increased at 24h to 72h of hypoxia. These data are included in Supplementary Fig. 1c of the revised manuscript.

7. The images of sprouting angiogenesis look quite marginal. Where these experiments blinded?

Answer: During the first round of evaluation, the experiments were not blinded. To overcome this potential source of error, the data was re-evaluated by a different person under encrypted conditions. However, as seen in Fig. 5a, b for the Reviewer, re-evaluation did not significantly change the experimental outcomes.

8. The experimental evidence of effects on angiogenesis and activation tags would be reassuring with an overexpression strategy, under basal and stimulated conditions.

Answer: We fully agree with the reviewer and overexpressed full length GATA6-AS in HUVECs by lentiviral transduction. As shown in Fig. 6a for the Reviewer, this led to an approximately 2 fold increased expression of GATA6-AS. As anticipated, GATA6-AS overexpression led to a de-repression of the LOXL2 target gene PTGS2 (see Fig. 6b for the Reviewer) and tend to augment in vitro sprouting (see Fig. 6c for the Reviewer).

9. The GAPMERs, how do they affect the different transcripts of the GATA6-AS1 transcript locus?

Answer: As extensively outlined in response to point 2, our RT-PCR data shows that the hypoxia-induced transcript variant GATA6-AS-001, as well alternatively spliced isoforms, are all downregulated by GATA6-AS GapmeR transfection (Fig. 2 a-e for the Reviewer). Importantly, although the annotated transcript variants GATA6-AS-003 and -004 do not harbor the GapmeR target site, we detect a significant down-regulation of transcript variants covering spliced transcripts

containing exons 2 and 5 (Fig. 2b-e for the Reviewer). This indicates that the exons 4 and 4' may not be correctly annotated.

10. The axis looks as though it is conserved in mouse. Although this requires some thought and validation, the authors should consider whether this is A. relevant and correct and B. A conserved function of the lncRNA.

Answer: We thank the reviewer for this comment and elucidated the expression of a putative mouse GATA6-AS orthologue. However, we were unable to identify a sequence homologue of the human GATA6-AS transcript. Next we sought for locus conservation and determined the expression of a mouse transcript in the region of interest. Indeed, several putative murine transcripts are annotated in the UCSC genome browser. However, no transcript was detectable in cultured mouse endothelial cells (Supplementary Fig. 3a and 3c) and in normoxic or ischemic muscle tissue from adult mice (Supplementary Fig. 3d). Interestingly, the GATA6-AS locus was active during embryonic development (Supplementary Fig. 3b), suggesting a potential developmental role of the locus in mice. However, genetic mouse models to study the role of GATA6-AS in development were precluded by the fact that the Gata6-AS transcription unit encompasses the promoter and other regulatory regions of Gata6. Therefore, CRISPR/Cas9-mediated excision would eliminate transcription of the essential Gata6 gene. The lack of expression in endothelial cells or in adult muscle tissue and the absence of sequence conservation additionally excludes GapmeR or short hairpin approaches to study the effect of the mouse transcript in a hind limb ischemia model *in vivo*.

To circumvent this limitation, we used a previously described xenograft model allowing us to address the *in vivo* function of transplanted human endothelial cells (Kaluza et al, 2011, EMBO J. 30:4142-56; Laib et al., 2009, Nat Protoc. 4:1202-15): We showed that transplantation of GATA6-AS GapmeR treated human endothelial cells led to the generation of more human vessels in matrigel plugs *in vivo*, whereas endogenous murine vessels were not changed (Fig. 2f, g, h, of the revised manuscript). Moreover, vessels formed by GATA6-AS-silenced endothelial cells were perfused as evidenced by the detection of erythrocytes in the lumen of these vessels, which indicates that the formed vessels are functionally mature (Supplementary Fig. 2e of the revised manuscript).

Response to Reviewer #2:

The manuscript is of interest and well describes the complexity of the mechanism of action of a lncRNA identified in hypoxic conditions, which influences more than one biological process at various regulation levels.

Criticisms:

1. RNA seq was conducted in HUVEC cells in hypoxic conditions; RIP experiments were conducted in HUVEC in normoxic conditions; CLIP and pull-down experiments in HeLa cells in normoxic conditions. While all these experiments prove that the lncRNA can bind loxl2 in all these instances, it does not prove its specific action in hypoxic conditions in HUVEC. Thus, the authors should perform a RIP for loxl2 in HUVEC cells in hypoxic conditions and compare the enrichment for the lncRNA between normoxic and hypoxic conditions. If binding is specific of hypoxic conditions, there should be an increased binding between the epigenetic protein and the lncRNA.

Answer: *To answer the reviewer's question, we performed LOXL2 RIP also under hypoxic conditions and assayed for the enrichment of GATA6-AS by RT-qPCR. As shown in Supplementary Fig. 5d, hypoxic conditions indeed increased the association between LOXL2 and the lncRNA.*

2. The authors silence both the lncRNA and loxl2 and show with RNAseq and ChIP seq for H3K4me3 that there is a negative correlation between the expression of loxl2 target genes and the silencing of the two genes. The specificity of the data could be improved by overexpressing the lncRNA in HUVEC in normoxia, comparing then RNAseq and ChIPseq results with those obtained treating cells with loxl2 siRNA. Are the genes modified by loxl2 silencing direct targets of this epigenetic factors? This needs to be clarified.

Answer: *We fully agree with the reviewer and overexpressed full length GATA6-AS in HUVECs by lentiviral transduction. As shown in Fig. 6a for the Reviewer, this led to an approximately 2 fold increased expression of GATA6-AS. In a next step, we measured the expression of the target gene PTGS2, which did reflect the results obtained after LOXL2 silencing (Fig. 6b for the Reviewer).*

To answer the question whether the genes regulated by LOXL2 silencing are also direct targets of LOXL2, we tried to perform LOXL2 ChIP experiments. Although we successfully immunoprecipitated LOXL2, we could not obtain co-precipitated DNA, thereby not allowing us to perform PCR analysis. The lack of direct interaction of LOXL2 with DNA may be explained by its mode of action: In our model, LOXL2 is (likely transiently) associated with H3K4me3 marks to deaminate these residues and does not interact with DNA.

We have ample data supporting that GATA6-AS regulates LOXL2 and gene expression by this mechanism: First, GATA6-AS is directly associated with LOXL2 and H3K4me3 ChIP analyses showed reduced tri-methylation states upon silencing of GATA6-AS (see Fig. 4c and 6e). Second, the promoter regions of PTGS2 and POSTN exhibit pronounced H3K4me3 signatures (see Supplementary Fig. 7h of the revised manuscript) and are therefore responsive to LOXL2-mediated gene regulation. Taken together, these results strongly argue for LOXL2 indirectly acting on gene expression by regulating the epigenetic control of the described target genes.

3. Most of the study is in vitro; the value of the manuscript would increase significantly if the authors could confirm some of the data also in vivo, by RIP for instance, in models of ischemia.

Answer: We thank the reviewer for this comment and elucidated the expression of a putative mouse GATA6-AS orthologue. However, we were unable to identify a sequence homologue of the human GATA6-AS transcript. Next we sought for locus conservation and determined the expression of a mouse transcript in the region of interest. Indeed, several putative murine transcripts are annotated in the UCSC genome browser. However, no transcript was detectable in cultured mouse endothelial cells (see Supplementary Fig. 3a and 3c) and in normoxic or ischemic muscle tissue from adult mice (see Supplementary Fig. 3d). Interestingly, the GATA6-AS locus was active during embryonic development (see Supplementary Fig. 3b), suggesting a potential developmental role of the locus in mice. However, genetic mouse models to study the role of GATA6-AS in development were precluded by the fact that the Gata6-AS transcription unit encompasses the promoter and other regulatory regions of Gata6. Therefore, CRISPR/Cas9-mediated excision would eliminate transcription of the essential Gata6 gene. The lack of expression in endothelial cells or in adult muscle tissue and the absence of sequence conservation additionally excludes GapmeR or short hairpin approaches to study the effect of the mouse transcript in a hind limb ischemia model in vivo.

To circumvent this limitation, we used a previously described xenograft model allowing us to address the in vivo function of transplanted human endothelial cells (Kaluza et al, 2011, EMBO J. 30:4142-56; Laib et al., 2009, Nat Protoc. 4:1202-15). We showed that transplantation of GATA6-AS GapmeR treated human endothelial cells led to the generation of more human vessels in matrigel plugs in vivo, whereas endogenous murine vessels were not changed (Fig. 2f, g, h of the revised manuscript). Moreover, vessels formed by GATA6-AS-silenced endothelial cells were perfused as evidenced by the detection of erythrocytes in the lumen of these vessels, which indicates that the formed vessels are functionally mature (see Supplementary Fig. 2e of the revised manuscript).

Minor:

1. Fig. 1 shows that the silencing of the lncRNA causes a significant reduction of angiogenic sprouting in the normoxic condition, suggesting a role of this lncRNA in angiogenesis that is not limited to hypoxia. Can the author speculate on this aspect? Is the effect increased in hypoxic condition?

Answer: According to our data, GATA6-AS is expressed at basal levels under normoxia, implying an - at least- partial repression of LOXL2 under these conditions. Silencing of GATA6-AS is expected to prevent the inactivation of LOXL2, which in turn regulates different target genes and eventually results in a changed sprouting behavior. We have evidence that hypoxia drives GATA6-AS expression independent on HIF1 α but in an Akt-dependent manner (Fig. 1e and Supplementary Fig. 1d, e, f of the revised manuscript). Which stimuli, besides hypoxia, change GATA6-AS expression cannot readily be answered, however, we have preliminary evidence that GATA6-AS is induced by shear stress and in silico promoter analysis revealed numerous potential transcription factor binding sites for GATA6-AS (see Fig. 3 for the Reviewer).

To answer the question whether sprouting defects are increased under hypoxia, we attempted to perform in vitro sprouting assays under hypoxic conditions, however, the exposure of spheroids to hypoxia reduced sprouting (Fig. 7 for the Reviewer), likely because the limited diffusion of oxygen already induces hypoxia in the spheroids under normoxic conditions and additional hypoxia may result in suppressive effects. Therefore, this model was not suitable to address the question of the

reviewer. However, we have obtained additional data showing that hypoxia leads to an augmented binding of GATA6-AS to LOXL2 (Supplementary Fig. 5d).

2. Can the author confirm the nuclear localization of this lncRNA with another technique? For instance, FISH.

Answer: Given that FISH would provide another layer of evidence for the nuclear localization of our lncRNA, we tried to establish this technique for GATA6-AS. Whereas our positive control targeting MALAT1 showed clear results, the specificity of the GATA6-AS signal was not conclusive, most probably due to the relatively low expression of GATA6-AS. However, as an alternative, we used a different cell fractionation protocol, described by Méndez and Stillman, to separate chromatin and cytoplasmic fractions (Méndez and Stillman, 2000, *Mol Cell Biol.* 20:8602-12). Using this alternative fractionation protocol, we found GATA6-AS to be predominantly associated with the chromatin fraction which is in line with its molecular mechanism of LOXL2-mediated histone modification (see Fig. 8 for the Reviewer).

3. The authors state that this lncRNA does not work in "cis". Have they tried to determine this in hypoxic conditions? Please explain better in the text.

Answer: As suggested by the reviewer, we assayed for a putative cis-regulatory function of GATA6-AS under hypoxic conditions by analyzing GATA6 mRNA and protein levels. Consistent with our previous results, no changes in GATA6 levels were detected. This new data is provided in the revised manuscript as Supplementary Fig. 4a, b. Accordingly, the text was changed and reads now:
Results: "However, a cis-regulatory function of GATA6-AS on the adjacent transcription factor GATA6 itself was not detectable in GATA6-AS-silenced normoxic or hypoxic endothelial cells (Supplementary Fig. 4a, b)."

Reviewer #3:

The manuscript from Neumann et al show how the lncRNA GATA6-AS interacts with LOXL2 acting as a negative regulator of LOXL2 activity in endothelial cells under hypoxia conditions. Through this interaction the lncRNA GATA6-AS is able to regulate endothelial gene expression. Although the results are very interesting, the model proposed has not been completely demonstrated by the current evidences and further experiments are required.

Major concerns

The authors showed how under hypoxia there is an up-regulation of several lncRNA and they focus their attention to the lncRNA GATA6-AS.

Figure 1d-f: All the experiments shown in these panels have to be done under hypoxia conditions, the situation in which the authors claim that the lncRNA GATA6-AS is induced and control endothelial gene expression.

Comment: This situation happens throughout the manuscript, most of the experiments shown have been done under normoxia conditions. For the reviewer is then not clear if the lncRNA GATA6-AS is also expressed under normoxia and if so is its function in normoxia different than under hypoxia?

Answer: *As requested by the reviewer, we repeated the mentioned experiments under hypoxic conditions. The subcellular fractionation revealed that -also upon hypoxia- GATA6-AS is predominantly localized to the nucleus, as shown in Supplementary Fig. 1g of the revised manuscript. To answer the question whether sprouting defects are increased under hypoxia, we attempt to performed in vitro sprouting assays under hypoxic conditions, however, the exposure of spheroids to hypoxia already reduced sprouting (Fig. 7 for the Reviewer), likely because the limited diffusion of oxygen already induces hypoxia in the spheroids under normoxic conditions and additional hypoxia may result in suppressive effects.*

However, we performed LOXL2 RIP under hypoxia and assayed for the enrichment of GATA6-AS by RT-qPCR. As shown in Supplementary Fig. 5d, hypoxic conditions indeed increased the association between LOXL2 and GATA6-AS.

Gene Ontology analysis from the experiment showed in figure S4C can be indicated here. GO terms have to be related with the pro-angiogenic function suggested for the lncRNA GATA6-AS.

Answer: *According to the reviewer's suggestion, we performed two pathway analyses (<http://metascape.org/gp/index.html#/main/step1>) using different combinations of regulated genes (see Fig. 9a, b for the Reviewer). First, we analyzed all significantly regulated genes upon silencing of GATA6-AS that were ≥ 1.2 fold up- or down-regulated (see Fig. 9a for the Reviewer). GATA6-AS affects pathways that are involved in cell communication such as Gap junctions, cell-to-cell junctions, and cell-to-cell communication; all of them are well-known to be key for endothelial cell biology and particularly play a role in EndMT.*

Second, when selecting genes which are inversely regulated upon GATA6-AS or LOXL2 silencing (see Fig. 9b for the Reviewer), pathways involved in tissue remodeling, endothelial development, and cellular response to stress were significantly affected. Importantly, these genes are most likely controlled by our proposed model of GATA6-AS-LOXL2-mediated gene regulation.

Figure S2: Here the authors suggest that lncRNA GATA6-AS is not acting in cis, however again this experiment has to be done under hypoxia conditions

Answer: *As suggested by the reviewer, we assayed for a putative cis-regulatory function of GATA6-AS under hypoxic conditions by analyzing GATA6 mRNA and protein levels. Consistent with our previous results, no changes in GATA6 levels were detected. This new data is provided in the revised manuscript as Supplementary Fig. 4a, b.*

Figure 3: The authors demonstrate the interaction between lncRNA GATA6-AS and LOXL2 in HeLa lysates and in HUVEC. In panel a the authors should show LOXL2 levels (input) in the control (scramble) and GATA6-AS conditions. In panel b the authors should show by western-blot His-LOXL2 expression.

Answer: *We regret any possible misinterpretations caused by the figure layout, however, would like to state that the input of Fig. 4a refers to both conditions, control- and GATA6-AS selection. To remove ambiguity, the figure legend was revised and now reads: "(a) Total cell lysate was prepared from HeLa cells and 5% was taken as input. The remaining lysate was distributed between GATA6-AS or scramble control antisense affinity selections and co-purified proteins were assayed for LOXL2 by western blotting."*

Fig. 3b and 3c of the initially submitted manuscript show the association of LOXL2 with GATA6-AS by RNA immunoprecipitation, relying exclusively on endogenous LOXL2: The input lane of old Fig. 3b shows initial LOXL2 levels from HUVEC cell lysate. Old Fig. 3c instead shows enrichment of GATA6-AS measured by RT-qPCR in immunoprecipitated samples. In order to avoid misunderstandings, we combined the immunoprecipitation and RT-PCR results as revised Fig. 4b and changed the figure legend accordingly.

Figure 4. It is known the LOXL2 affects cellular proliferation and cell death, the effect showed by the authors in spheroid sprouting assays can be related with cell proliferation rather than angiogenesis. Moreover, based on gene expression profile if LOXL2 and lncRNA GATA6-AS share sets of inversely regulated genes, and lncRNA GATA6-AS is pro-angiogenic, LOXL2 should be anti-angiogenic. If this is the case, why the absence of LOXL2 results in a sprouting phenotype? This is the same phenotype shown when lncRNA GATA6-AS is knocked down.

Answer: *We determined the effects of GATA6-AS on cellular proliferation and apoptosis and observed a modest reduction of apoptosis and a slight increase in proliferation. However, these effects were not significant (see Supplementary Fig. 2c/d).*

We agree with the reviewer that, based on the set of inversely regulated genes of GATA6-AS and LOXL2, LOXL2 silencing should be expected to increase in vitro sprouting angiogenesis. However, we could show that silencing of GATA6-AS did not change intracellular or extracellular levels of LOXL2, which is in clear contrast to direct silencing of LOXL2 (see Fig. 5d of the revised manuscript). This raises the question whether the lack of extracellular LOXL2 is responsible for mediating the observed sprouting defect after silencing of LOXL2.

In order to analyze the role of extracellular LOXL2 in sprouting angiogenesis in more detail, we silenced LOXL2 in endothelial cells and determined whether treatment with supernatant from control- or LOXL2-silenced cells restored sprouting behavior. Indeed, supernatants of control cells, but not from LOXL2-silenced endothelial cells, rescued the angiogenesis defect in LOXL2 siRNA transfected cells (Supplementary Fig. 6). Taken together, these results demonstrate that 1. GATA6-AS silencing does not alter extracellular LOXL2 levels and 2. the LOXL2 sprouting phenotype is -at least in part- driven by an extracellular depletion of LOXL2 and thus cannot not directly be compared to conditions of GATA6-AS silencing.

Figure 5. In this figure the authors try to demonstrate how the interaction between LOXL2 and lncRNA GATA6-AS interferes with LOXL2 activity.

Fig5a: P value? Is this decrease significant? Again, this reduction in the global levels of H3K4me3 have to be shown under hypoxia. Having say that, sometimes is difficult to observe changes in histone marks at global levels, probably a limited number of genes will be under the control of LOXL2-GATA6-AS.

Loading control must be total histone H3 instead of tubulin.

Answer: *For proper statistical analysis, we increased the number of experiments analyzing the change in global H3K4me3 levels to n=6 and found that reduction of lysine trimethylation was significant upon silencing of GATA6-AS. This new data is shown in revised Fig. 6a. In addition, as requested by the reviewer, we repeated these experiments under hypoxic conditions, now shown in Supplementary Fig. 7b, c, and found that H3K4me3 levels are reduced by GATA6-AS silencing. Finally, equal loading of the western blots was controlled by total histone H3 levels in normoxic and hypoxic conditions.*

Fig5e: P-value? For the reviewer is not clear if this decrease is significant. Is the CHIP-PCR corrected by the total amount of H3? Irrelevant genomic regions have to be shown. Also it would be nice to show a control positive region, a LOXL2 target promoter (End-Cadherin maybe?).

LOXL2 CHIP in these suggested regions is also a must. Again, the effects should be increase under hypoxia conditions.

Answer: *We understand the reviewer's concerns and increased the number of repeats for the H3K4me3 CHIP to n=10. Importantly, CHIP efficiencies on the PTGS2 and POSTN promoters are now significantly reduced. In addition, as suggested by the reviewer, we included irrelevant genomic regions, lacking H3K4me3 marks, as negative controls. These new data are shown in Fig. 6e of the revised manuscript.*

We additionally tried to perform LOXL2 CHIP experiments. Although we successfully immunoprecipitated LOXL2, we could not obtain sufficient amounts of DNA for subsequent analysis (neither under normoxic nor hypoxic conditions). The lack of direct interaction of LOXL2 with DNA may be explained by its mode of action: In our model, LOXL2 is (likely transiently) associated with H3K4me3 marks to deaminate these residues and does not interact with DNA. To put this into a biochemical context, the Michaelis constant (K_M) of LOXL2 on different substrates was determined to be ~1mM (Rodriguez et al., 2010, J Biol Chem. 285:20964-74) and therefore can be considered below-

average. In other words, the chemical equilibrium of this enzymatic reaction is in the dissociated state therefore likely not allowing the co-purification of its substrates that are associated with the DNA.

However, we have ample data supporting that GATA6-AS regulates LOXL2 and gene expression by this mechanism: First, GATA6-AS is directly associated with LOXL2 and H3K4me3 ChIP analyses showed reduced tri-methylation states upon silencing of GATA6-AS (see Fig. 4c and 6e). Second, the promoter regions of PTGS2 and POSTN exhibit pronounced H3K4me3 signatures (see Supplementary Fig. 7h of the revised manuscript) and are therefore responsive to LOXL2-mediated gene regulation. Taken together, these results strongly argue for LOXL2 indirectly acting on gene expression by regulating the epigenetic control of the described target genes.

Moreover, by repeating LOXL2 RNA immunoprecipitation reactions under hypoxia, we were able to show that hypoxic conditions indeed increased the association of LOXL2 with GATA6-AS, as shown in Supplementary Fig. 5d.

To demonstrate that LOXL2 activity decreases in vitro assays have to be done. The authors can use the Amplex-Red kit that measures the levels of H₂O₂ released upon LOXL2 reaction. Recombinant LOXL2 protein + lncRNA GATA6-AS + substrate (H3K4me3 peptides) can be used. Also by western blot the authors can check in nucleosomes incubated with LOXL2 in the presence or absence of lncRNA GATA6-AS histone marks levels.

Answer: We followed the reviewer's suggestion and tried to establish in vitro deamination assays to elucidate the inhibitory function of GATA6-AS on LOXL2. To this end, recombinant LOXL2 (R&D Systems) was incubated with purified nucleosomes in the presence or absence of in vitro transcribed and purified full length GATA6-AS. The catalytic activity of LOXL2, reflected by the generation of H₂O₂, was determined using Amplex Red assays. As shown in Fig. 10a for the Reviewer, the data indicates that the recombinant LOXL2 was not catalytically active in our experimental setup and this way we were not able to show direct evidence for a GATA6-AS-mediated inhibition of LOXL2. Analytical sensitivity of the Amplex Red reaction was controlled by a serial dilution of H₂O₂ as shown in Fig. 10b for the Reviewer. Nevertheless, as shown in improved Figure 6a of the revised manuscript (sample size increased to n=6), we clearly could detect a significant reduction of global H3K4me3 levels upon silencing of GATA6-AS in HUVEC cell culture.

Minor concern

Herranz et al, Febs J must be cited, this is the paper that describes the role of LOXL2 as a histone deaminase protein.

Answer: We thank the reviewer for this reference and cite the work of Herranz et al. in the revised manuscript. The text now reads:

"Nuclear LOXL2 in contrast is known to act as a transcription co-repressor by specifically deaminating trimethylated lysine 4 of histone H3 (H3K4me3), a chromatin signature for transcriptional activation^{35,36}."

Reviewer #4 :

In this study, the authors identified 64 hypoxia-responsive long non-coding RNAs (lncRNAs) in HUVEC cells. Among them, GATA6-AS is an uncharacterized transcript that showed consistent response (up-regulation) to hypoxia treatment for either 12h or 24h. The authors go on further to characterize the biological function of GATA6-AS and demonstrate that GATA6-AS localized mainly in the nuclear and played pro-angiogenic functions in HUVEC cells. By performing GATA6-AS affinity selection and mass spectrometry in HeLa cells, the authors identified multiple protein binding partners of GATA6-AS in which Lysyl oxidase-like 2 (LOXL2) was among the top enriched proteins. The authors then performed GATA6-AS affinity selection followed by western blotting, RNA immunoprecipitation (RIP), in vitro His-tag pulldown assay followed by northern blotting and individual-nucleotide resolution crosslinking and immunoprecipitation (iCLIP) to confirm the interaction between GATA6-AS and LOXL2. Two LOXL2 iCLIP-tags were identified on GATA6-AS. It appears that GATA6-AS silencing did not affect intra- or extracellular LOXL2 levels although there's a direct binding between them. Given LOXL2 has been reported to play nuclear function as a transcription co-repressor through deaminating H3K4me3, the authors examined the expression of H3K4me3 in HUVECs and found modest reduction (17%) of H3K4me3 after GATA6-AS silencing. The authors then performed exon arrays to study endothelial gene expression profile in either GATA6-AS- or LOXL2-silenced cells. And they found an inversely regulated gene expression pattern between GATA6-AS- or LOXL2-silenced groups. Finally, the authors picked one from each group (COX-2 and POSTN) for H3K4me3 ChIP-PCR analysis and found GATA6-AS silencing reduced H3K4me3 ChIP efficiencies on the promoter regions of COX-2 and POSTN. Taken together, the authors drove the conclusion that GATA6-AS is a hypoxia-responsive transcript, and that GATA6-AS epigenetically regulates endothelial gene expression and plays pro-angiogenic roles through interacting with LOXL2.

Overall, this is a study consisting of extensive amount of work. However the possible mechanism has modest novelty since LOXL2 has been well characterized in hypoxic response and angiogenic function in endothelial cells, so is LOXL2's transcriptional repressing function through H3K4me3. In addition, several observations presented here are not convincing enough to support the conclusions. There are a number of points that need to be addressed as outlined below.

1. The subcellular localization of GATA6-AS in HUVEC cells was determined by cellular fractionation in Fig. 1D. It would be more convincing if the nuclear localization of GATA6-AS can be confirmed by RNA-FISH in HUVECs under both normoxic and hypoxic condition.

Answer: *We agree with the reviewer's opinion that FISH would add another layer of evidence for the subcellular localization of GATA6-AS and we therefore tried to establish this technique for GATA6-AS. Whereas our positive control, MALAT1, showed clear results, the specificity of the GATA6-AS signal was not conclusive, most probably due to the relatively low expression of GATA6-AS. However, we repeated the cellular fractionation under hypoxic conditions and likewise found GATA6-AS to be nuclear localized. This data is included as Supplementary Fig. 1g in the revised manuscript. As an additional proof, we used a different cell fractionation protocol, described by Méndez and Stillman, to separate chromatin and cytoplasmic fractions (Méndez and Stillman, 2000, Mol Cell Biol. 20:8602-12). Using this alternative fractionation protocol, we found GATA6-AS to be predominantly associated with the chromatin fraction which is in line with our proposed molecular mechanism of GATA6-AS-LOXL2-mediated histone modification (see Fig. 8 for the Reviewer).*

2. The authors studied the biological functions of hypoxia-responsive GATA6-AS in HUVEC cells. It would be nice to see if GATA6-AS also plays pro-angiogenic effect in in vivo settings, such as hind limb ischemia model, retinal hypoxia, etc. In vivo studies would have more clinical impact for the current study.

Answer: We thank the reviewer for this comment and elucidated the expression of a putative mouse GATA6-AS orthologue. However, we were unable to identify a sequence homologue of the human GATA6-AS transcript. Next we sought for locus conservation and determined the expression of a mouse transcript in the region of interest. Indeed, several putative murine transcripts are annotated in the UCSC genome browser. However, no transcript was detectable in cultured mouse endothelial cells (Supplementary Fig. 3a and 3c) and in normoxic or ischemic muscle tissue from adult mice (Supplementary Fig. 3d). Interestingly, the GATA6-AS locus was active during embryonic development (Supplementary Fig. 3b), suggesting a potential developmental role of the locus in mice. However, genetic mouse models to study the role of GATA6-AS in development were precluded by the fact that the *Gata6-AS* transcription unit encompasses the promoter and other regulatory regions of *Gata6*. Therefore, CRISPR/Cas9-mediated excision would eliminate transcription of the essential *Gata6* gene. The lack of expression in endothelial cells or in adult muscle tissue and the absence of sequence conservation additionally excludes GapmeR or short hairpin approaches to study the effect of the mouse transcript in a hind limb ischemia model in vivo.

To circumvent this limitation, we used a previously described xenograft model allowing us to address the in vivo function of transplanted human endothelial cells (Kaluza et al, 2011, EMBO J. 30:4142-56; Laib et al., 2009, Nat Protoc. 4:1202-15). We showed that transplantation of GATA6-AS GapmeR treated human endothelial cells led to the generation of more human vessels in matrigel plugs in vivo, whereas endogenous murine vessels were not changed (Fig. 2f, g, h of the revised manuscript). Moreover, vessels formed by GATA6-AS-silenced endothelial cells were perfused as evidenced by the detection of erythrocytes in the lumen of these vessels, which indicates that the formed vessels are functionally mature (see Supplementary Fig. 2e of the revised manuscript).

3. GATA6-AS was identified as hypoxia-responsive transcript in endothelial cell, also the main conclusion of the current study is that GATA6-AR regulates endothelial gene expression. However, the authors switched to HeLa cells for the antisense affinity selection when trying to identify protein binding partners of GATA-AR (Fig. 2d) and repeated the affinity selection of GATA6-AS also in HeLa cell lysates (Fig. 3a). For the following RIP and iCLIP assays, the authors switched back to HUVECs. It is a somewhat strange experiment design. It would be more reasonable if the affinity selection was conducted using HUVECs.

Answer: We follow the argument of the reviewer, however, using an easy to handle cell line (instead of primary cultured endothelial cells) allowed us to maximize the protein yield of the affinity selection which is a prerequisite for reliable mass spectrometry results. We want to emphasize that GATA6-AS is well expressed and induced by hypoxia in HeLa cells, suggesting that the biological context is reliable for detecting RNA-protein interactions. This assumption is also confirmed by our subsequent validation studies in HUVECs, demonstrating the interaction between GATA6-AS and LOXL2 by RNA immunoprecipitation (Fig. 4b) as well as by LOXL2 iCLIP sequencing (Fig.4d bottom).

4. The authors employed four independent assays to confirm the interaction between GATA6-AS and LOXL2, which is very impressive. However it would be more convincing if the nuclear localization of LOXL2 can be confirmed in HUVEC cells under normal and hypoxic condition, given it is a frequently raised question that how nuclear localized lncRNAs bind cytosolic localized RNA/proteins. Also it would be interesting to know whether the interaction between GATA6-AS and LOXL2 would have any difference under physiological and pathological conditions.

Answer: *We followed the reviewer's suggestion and analyzed the subcellular distribution of LOXL2 under normoxic and hypoxic conditions by cell fractionation and LOXL2 western blotting. The results are shown as Supplementary Fig. 7a in the revised manuscript. Clearly, LOXL2 localizes to both compartments, the nucleus and the cytoplasm. Of note, the nuclear LOXL2 level seems to be slightly increased upon hypoxia. In order to answer the question whether the interaction of GATA6-AS and LOXL2 would be different under pathological conditions, we performed LOXL2 RIP under hypoxia and assayed for the enrichment of GATA6-AS by RT-qPCR. As shown in Supplementary Fig. 5d, hypoxic conditions indeed increased the association between LOXL2 and GATA6-AS.*

5. Fig. 4C needs better western blot image with quantification results. Tubulin looks uneven in the current blot.

Answer: *We agree with the reviewer and repeated western blotting for the loading control tubulin. Based on these new results, nuclear LOXL2 levels were also quantified. The new western blot and the quantification of the bands are now shown as Fig. 5d in the revised manuscript.*

6. I wonder whether a p value has been calculated for H3K4me3 quantification result presented in Fig. 5a. It appears that the differences between LNA Ctl and LNA GATA6-AS groups are not significant.

Answer: *We understand the reviewer's concerns and increased the number of replicates for the analysis of global H3K4me3 levels to n=6, leading to statistically significant results. The improved data is now shown as Fig. 6a in the revised manuscript.*

7. The ChIP-PCR analysis results (Fig. 5e) are not convincing either (no P value?). Also sample size (n=3) is highly recommended for gene expression profiling studies (Fig. S4c).

Answer: *In order evaluate our data statistically, we increased the sample size of the ChIP-PCR analysis to n=10 and showed that GATA6-AS silencing significantly reduced H3K4me3 marks at the promoters of the target genes.*

With respect to an increased sample size for the gene profiling studies (which are associated with high costs), we would like to stress, that these assays were used to uncover candidate genes for the model of inverse gene regulation, which were subsequently confirmed by RT-qPCR. As shown in Fig. 6d, our analyzed subset angiogenesis-related genes turned out to be significantly regulated in both conditions, LOXL2 and GATA6-AS silencing. Finally, our mechanistic findings are strengthened by revised Fig. 6e, now showing significantly reduced H3K3me4 ChIP efficiencies on two selected target

genes upon silencing of GATA6-AS. Taken this into account, we hope that the reviewer agrees that additional micro arrays would not greatly add on the current data.

REVIEWERS' COMMENTS:

Reviewer #1 (Remarks to the Author):

The authors have made a solid and scientifically justified response to my comments. This manuscript is acceptable for publication, in my view. Andrew H Baker

Reviewer #2 (Remarks to the Author):

The authors have replied to the criticisms I raised in some cases with new experiments in others with honest and solid argumentations, which I hope will be included, though briefly, in the discussion.

I have no further requests.

Reviewer #3 (Remarks to the Author):

The authors have addressed my main questions satisfactorily. The manuscript has been improved with the new data. It is an interesting story worth of the high visibility offered by Nature Comm

Reviewer #4 (Remarks to the Author):

The authors have answered all my questions, and now the revised manuscript is suitable for acceptance for publication.